# NINscope, a versatile miniscope for multi-region circuit investigations

Andres de Groot[1], Bastijn JG van den Boom[1,2], Romano M van Genderen[3,4], Joris Coppens[1], John van Veldhuijzen[1], Joop Bos[1], Hugo Hoedemaker[1], Mario Negrello[3,4], Ingo Willuhn[1,2], Chris I De Zeeuw[1,4]*, Tycho M Hoogland[1,4]*

[1]Netherlands Institute for Neuroscience, Royal Netherlands Academy of Arts and Sciences, Amsterdam, Netherlands; [2]Department of Psychiatry, Amsterdam UMC, University of Amsterdam, Amsterdam, Netherlands; [3]Faculty of Applied Sciences, TU Delft, Delft, Netherlands; [4]Department of Neuroscience, Erasmus MC, Rotterdam, Netherlands

**Abstract** Miniaturized fluorescence microscopes (miniscopes) have been instrumental to monitor neural signals during unrestrained behavior and their open-source versions have made them affordable. Often, the footprint and weight of open-source miniscopes is sacrificed for added functionality. Here, we present NINscope: a light-weight miniscope with a small footprint that integrates a high-sensitivity image sensor, an inertial measurement unit and an LED driver for an external optogenetic probe. We use it to perform the first concurrent cellular resolution recordings from cerebellum and cerebral cortex in unrestrained mice, demonstrate its optogenetic stimulation capabilities to examine cerebello-cerebral or cortico-striatal connectivity, and replicate findings of action encoding in dorsal striatum. In combination with cross-platform acquisition and control software, our miniscope is a versatile addition to the expanding tool chest of open-source miniscopes that will increase access to multi-region circuit investigations during unrestrained behavior.

*For correspondence:
c.de.zeeuw@nin.knaw.nl (CIDZ);
hoogland@nin.knaw.nl (TMH)

Competing interests: The authors declare that no competing interests exist.

## Introduction

Cellular resolution imaging using miniaturized fluorescence microscopes (miniscopes) permits the monitoring of the topology of activity in brain circuits during unrestrained behaviors. While advances in electrophysiology now enable recordings from many thousands of neurons at once in awake animals (*Juavinett et al., 2018*; *Jun et al., 2017*), imaging approaches can sample the activity of individual neurons and retain information about how their activity is spatially distributed in a large network (*Terada et al., 2018*; *Stirman et al., 2016*; *Kim et al., 2016*). Often an anatomical substrate exists for clustered activity such as is the case in the cerebellum, where nearby Purkinje cells receive input from climbing fibers originating in adjacent neurons of the inferior olive brainstem nucleus (*Ruigrok, 2011*). Thus, imaging approaches can reveal how individual cells embedded in a larger network display coordinated activity during different stages of behavior or training (*Wagner et al., 2017*; *Heffley et al., 2018*; *Galiñanes et al., 2018*; *Giovannucci et al., 2017*; *Kostadinov et al., 2019*). Moreover, because of their ability to record in freely moving animals, miniscopes have been instrumental in uncovering neural activity patterns occurring during natural behaviors and related brain-states including social interactions (*Murugan et al., 2017*; *Remedios et al., 2017*; *Liang et al., 2018*; *Kingsbury et al., 2019*) or sleep (*Chen et al., 2018*; *Cox et al., 2016*) with fully intact vestibular input.

Open-source miniscopes are affordable tools to probe cellular activity in rodents (*Ghosh et al., 2011*; *Cai et al., 2016*) and birds (*Liberti et al., 2016*; *Liberti et al., 2017*) during unrestrained behavior and until recently have been limited to recordings from a single region, but see

*Gonzalez et al. (2019)*. To understand how brain circuits elicit behavior, it is necessary to probe multi-region interactions during either spontaneous or trained behaviors. If miniscopes are used to this end, they should be sufficiently light and compact to allow recordings from more than one site without compromising image quality, permit straight-forward behavioral tracking, and have the ability to drive circuits optogenetically. To address all these needs in one device, we have developed a versatile and compact miniscope (NINscope, named after the institute of origin) with a sensitive CMOS sensor, integrated inertial measurement unit (IMU) and an accurate LED driver for optogenetic actuation of other brain regions using a custom-made LED probe.

Leveraging the capabilities of NINscope, we demonstrate its ability to monitor functional interactions between the cerebellum and cortex in unrestrained mice wearing dual miniscopes, one of a number of possible dual-scope configurations with our miniscope. Complex spike activity in Purkinje cell dendrites correlated with neural activity measured in the cortex during periods of movement acceleration, in line with expectations based on previous anatomical and functional studies of cerebello-thalamo-cortical connectivity (*Badura et al., 2018*; *Bostan et al., 2013*; *Akkal et al., 2007*; *Hoover and Strick, 1999*; *Gao et al., 2018*; *Wagner et al., 2019*). Using NINscope's built-in optogenetic stimulation capabilities in conjunction with accelerometer read-out, we show that cerebellar stimulation elicits clearly discernible behavioral responses and activation of cortical neurons.

We further demonstrate the applicability of NINscope to image from neurons in the mouse dorsal striatum, a deep-brain region accessible only after implantation of a relay GRIN lens. The integrated accelerometer allowed us to identify cells in the dorsal striatum whose activity was exclusively modulated when mice make turns contralateral to the recording site, reiterating lateralization in the dorsal striatum and the role of these neurons in representing action space (*Klaus et al., 2017*; *Barbera et al., 2016*; *Cui et al., 2013*). Finally, we use NINscope to show that optogenetic activation of two different projection pathways results in differential modulation of activity in neurons of dorsal striatum.

Our validation experiments show that NINscope permits new types of recordings in unrestrained mice. We demonstrate the feasibility to record from two regions in the same mouse concurrently, to use a built-in LED driver in combination with a LED probe for optogenetic stimulation, and to parse behavioral states by the inclusion of an accelerometer. The integration of these components in a single device facilitates data acquisition and analysis. NINscope is an open-source project enabling others to build on its design and functionalities, thereby contributing to a growing range of open-source tools to study neural circuits during unrestrained behavior.

## Results

### NINscope design and functionality

NINscope (*Figure 1A–C*) distinguishes itself from other miniscopes by retaining a low weight and small form factor permitting dual site recordings, while offering a number of novel integrated features and cross-platform interoperability (for a comparison with other miniscopes see *Table 1*). A thinner optical emission filter and dichroic mirror (both 500 µm) were used and the emission filter (1 mm) was glued to a plano-convex lens using optical bonding glue (NOA81, Norland Products). Custom high-density interconnector (HDI) sensor and interface PCBs (10 by 10 mm, *Figure 1B*) were designed, fitted with electronic components using a pick-and-place machine (NeoDen4, NeoDen Tech, Hangzhou, China) and stacked to maintain a small footprint. This allowed for on-board integration of an IMU and multiple LED drivers including one for optogenetic actuation and two for driving excitation LEDs. The latter feature allows for future extension from single to dual excitation mode.

To allow optogenetic stimulation remote from the site of imaging, we developed a simple LED probe consisting of a Mill-Max connector, insulated enamel wires and an SMD LED with 402 case size, which was sealed with epoxy (*Figure 1D*).

The PYTHON480 sensor (ON Semiconductor) was chosen as a compact yet sensitive CMOS sensor (pixel size: 4.8 µm, dynamic range >59 dB) with modest power requirements. NINscope uses a 1.8 mm diameter GRIN lens (numerical aperture 0.55, #64–519, Edmund Optics) as objective and has a magnification of ~4.6 x (*Figure 1—figure supplement 1*). This gave us approximate field sizes of 786 by 502 µm, using a 752 × 480 pixels Region Of Interest (ROI). In software, we implemented the option to translate this ROI to cover the CMOS sensor area of 800 × 600 pixels (836 by 627 µm).

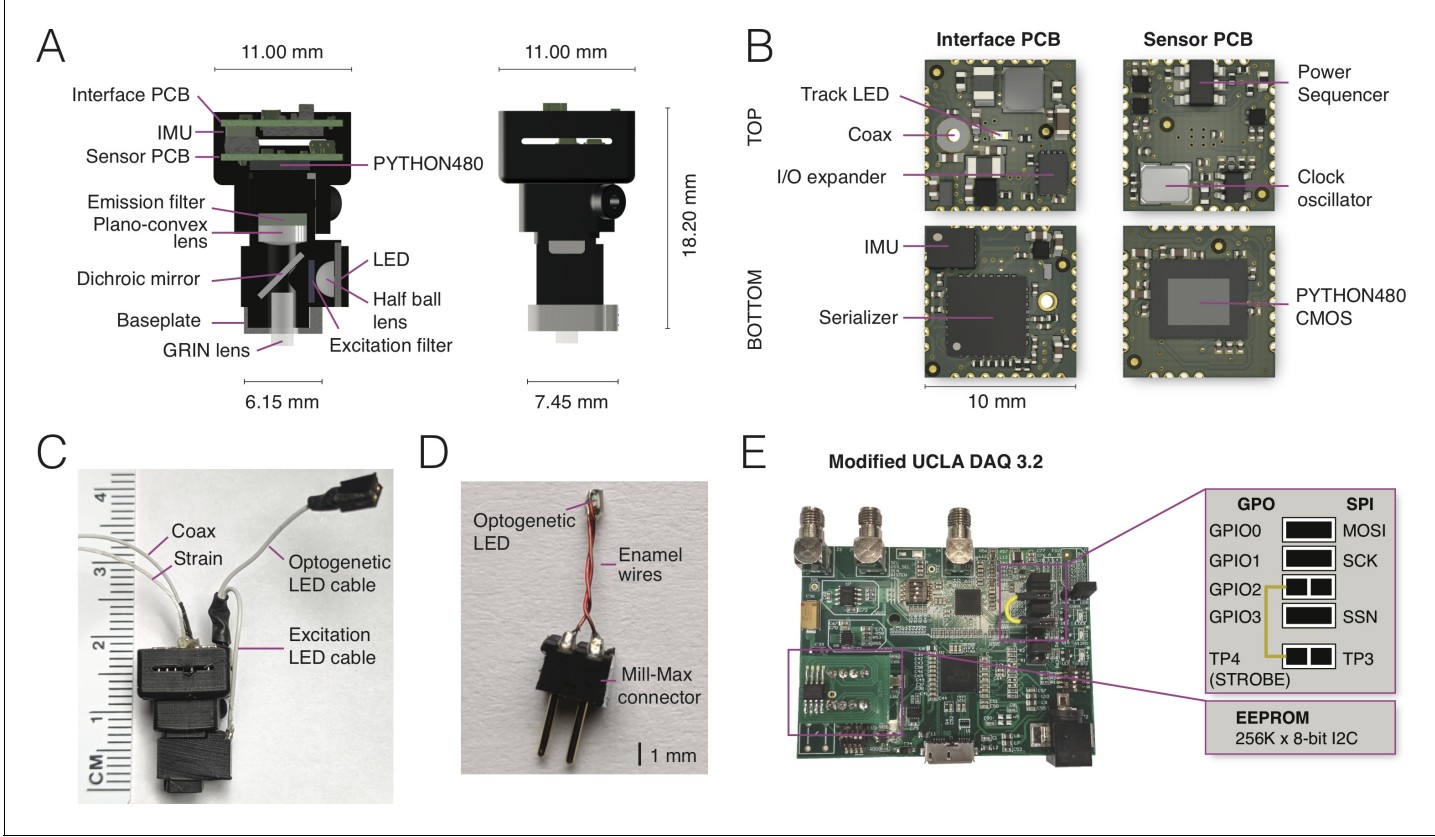

**Figure 1.** NINscope, a compact, light-weight and versatile miniscope. (**A**) Schematics of the NINscope with dimensions in mm. Two 10 by 10 mm HDI printed circuit boards (PCBs), one for interfacing with the data acquisition box (Interface PCB), the other containing the CMOS imaging sensor (Sensor PCB), are stacked and mounted in a 3D printed enclosure. Excitation light from an LED is collimated with a half ball lens, passes through an excitation filter and is reflected by the dichroic mirror onto the specimen. The emitted fluorescence is collected through the GRIN objective lens, and passes the dichroic and a plano-convex lens, which focuses an image onto the CMOS sensor. An emission filter is glued onto the plano-convex lens with optical bonding glue. (**B**) KiCad renders of the custom-built interface and sensor PCBs with top and bottom views. The interface PCB contains an inertial measurement unit (IMU) for measuring head acceleration and orientation, three LED drivers including one for optogenetic (strobe) control and two for excitation LEDs (one used), as well as a red tracking LED, the serializer, and the IO expander. The sensor PCB contains the PYTHON480 CMOS sensor, clock oscillator and a power sequencer, which provides the image sensor with the necessary voltages as well as their timing and sequence. (**C**) Photograph of NINscope with coax and strain cables, excitation LED and optogenetic LED cables. (**D**) Custom-built implantable LED probe for optogenetic stimulation that is connected to NINscope using the optogenetic LED cable. (**E**) The UCLA Miniscope DAQ card v3.2 was used with minor modifications including a 256 kB x 8-bit I2C EEPROM (STMicroelectronics) and a wired connection bridging general purpose input/output 2 (GPIO2) with test point 4 (TP4). The serial peripheral interface (SPI) signals: master output slave input (MOSI), serial clk (SCK) and slave select (SSN) are connected to GPIO0, GPIO1 and GPIO3 through jumpers.

The online version of this article includes the following figure supplement(s) for figure 1:

**Figure supplement 1.** NINscope optical design.
**Figure supplement 2.** NINscope baseplate.
**Figure supplement 3.** NINscope software.

Because of the widespread accessibility of the first generation UCLA Miniscope, we retained the data acquisition (DAQ V3.2) module of the UCLA Miniscope project with minor modifications that included an EEPROM to store a larger modified version of the latest Cypress EZ-USB FX3 firmware and a wired connection from general purpose input/output 2 (GPIO2) to test point 4 (TP4), allowing 1 ms timing accuracy for the optogenetic LED driver (*Figure 1E*). The firmware of the DAQ module was modified to enable serial control over optogenetic and excitation LED brightness, as well as gain, exposure and black level of the CMOS sensor.

The microscope housing was 3D printed (EnvisionTec Micro Plus Advantage printer, RCP30 M resin and Formlabs Form 2 printer, RS-F2-GPBK-04 black resin) to allow for rapid prototyping of various miniscope designs. Printing accuracy proved sufficient for our final design enabling us to keep

**Table 1.** Features of NINscope and other miniscopes.

An overview of the features of currently released open-source and commercially available (wide-field illumination) miniscopes. Opto-remote: ability to optogenetically stimulate outside of the imaging field. Opto in-field: ability to optogenetically stimulate in the imaging field. IMU: inertial measurement unit. eFocus: electric focusing using an electrowetting lens (EWL). 1: (*Cai et al., 2016*), 2: (*Shuman et al., 2020*), 3: (*Leman et al., 2018*), 4: (*Jacob et al., 2018*), 5: (*Zhang et al., 2019*), 6: (*Barbera et al., 2019*), 7: (*Gonzalez et al., 2019*). n.a. = data not available. *nVoke system. **Optogenetically Synchronized Fluorescence Microscope.

| Miniscope | Weight (g) | 3D printed | Cross platform? | Two scopes? | Opto-remote | Opto -in field | Microphone | IMU | eFocus |
|---|---|---|---|---|---|---|---|---|---|
| NINscope | 1.6 | yes | yes | yes | yes | no | no | yes | no |
| UCLA Miniscope v3[1] | 3.2 | no | no | no | no | no | no | no | no |
| UCLA Miniscope wireless[2] | 4. 5 | no | no | no | no | no | no | no | no |
| FinchScope[3] | 1.8 | no | no | n.a. | no | no | yes | no | no |
| CHEndoscope[4] | 4.5 | yes | no | no | no | no | no | no | no |
| miniScope Lin[5] | 2.4 | yes | yes | no | no | no | no | no | no |
| miniScope Lin Wireless[6] | scope: 3.9 battery:2.2–5 | yes | no | no | no | no | no | no | no |
| Miniscope Gonzalez et al.[7] | n.a. | no | n.a. | yes | no | no | no | no | no |
| Inscopix (commercial) | 1.8 and up | no | yes | n.a. | no | yes* | no | no | yes |
| Doric lenses (commercial) | 2.2, 3.0 excl. canula | no | no | no | no | yes** | no | no | yes |

the weight of the miniscope down to 1.6 grams (housing+optics+PCBs), while permitting the use of two miniscopes simultaneously on one mouse. The top half of NINscope has an enclosure for the sensor and interface PCBs to protect them from damage during unrestrained animal behavior. NINscope is secured onto a small-footprint baseplate (6.5 by 7.5 mm outer dimensions, *Figure 1—figure supplement 2*) with a set-screw. The lower half of the NINscope housing has a protrusion that fits in a notch in the baseplate for increased stability.

User-friendly software was developed in the Processing language (http://processing.org), allowing true cross-platform interoperability and control of experimental recordings (*Figure 1—figure supplement 3*). The option was included to record both in single or dual head (two miniscope) mode in combination with an additional USB webcam for video capture of behavior. Ring buffers were implemented in the software to avoid frame time delays and frame drops during acquisition. Timestamps of acquired frames were logged for post-hoc synchronization. We tested data acquisition on a number of computer configurations with various types of operating systems (Windows, MacOS, Linux), processors (i5, i7) and hard drives (SATA, SSD). The IMU accelerometer data can be displayed live during the miniscope recordings. In addition, optogenetic stimulation patterns and LED probe current can be adjusted through the interface providing integrated control of all aspects of the experiment (*Figure 1—figure supplement 3*). A more detailed description of the hardware and software is provided in the Materials and methods section. Design files and instructions on hardware assembly (for miniscope and LED probes), firmware programming and software installation can be found at our GitHub site: https://github.com/ninscope.

In order to validate our miniscope, we imaged under various recording and stimulation configurations, including single and dual scope modes across different brain regions. Behavior of animals was monitored with a USB webcam (*Figure 2A*), the miniscope tracking LED (*Figure 2B*), and the onboard accelerometer. In the example shown, Purkinje cells in lobule V of the cerebellum (AP: −6.4, ML: 0 mm) were transduced with GCaMP6f, their dendrites were imaged with NINscope and segmented using the CNMF-E suite (*Zhou et al., 2018*) (*Figure 2C,D*, *Figure 2—video 1*). For all analyses deconvolved calcium transients (neuron.C output obtained after running CNMF-E; see *Vogelstein et al., 2009*) were used to extract event times (*Figure 2C*). Climbing fibers derived from the inferior olive in the ventral medulla elicit complex spikes in Purkinje cells; they trigger calcium influx throughout the Purkinje cell dendrite arbor up to the brain surface and their firing frequency is stably maintained at around 1 Hz (*Ju et al., 2019*). The parasagittal arrangement of Purkinje cell

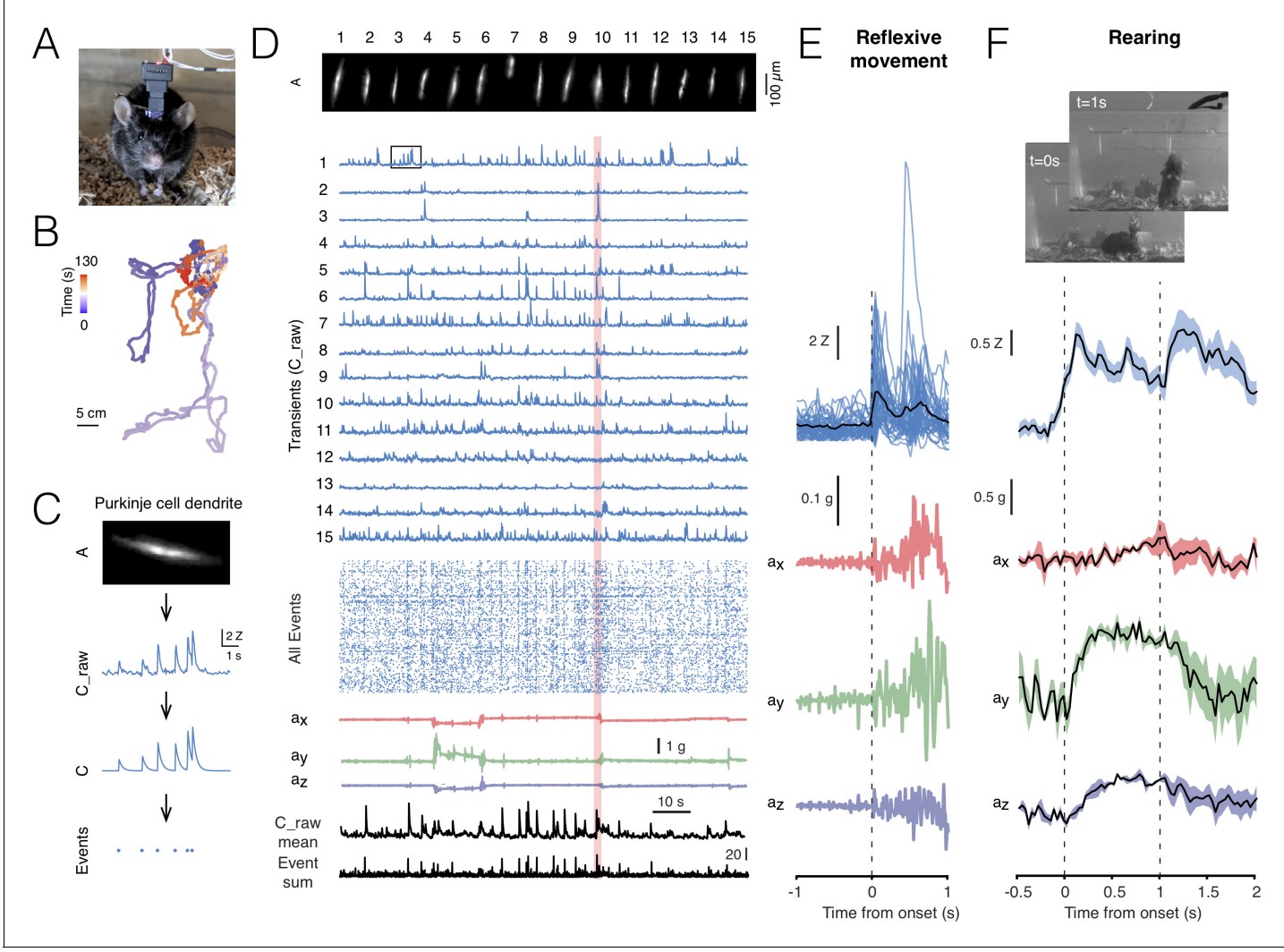

**Figure 2.** Cerebellar imaging with NINscope. (**A**) A mouse wearing a single NINscope mounted over lobule V of cerebellum, where Purkinje cells were selectively transduced with GCaMP6f. (**B**) Animal position can be extracted from the tracking LED when combined with concurrent webcam recordings. Colors represent the time from the start of recording and the track represents cage exploration over the course of ~2 min. (**C**) Spatial footprints of Purkinje cell dendrites (A) raw (C_raw) and deconvoluted signals (C) were extracted using CNMF-E (*Zhou et al., 2018*) after motion correction using NoRMCorre (*Pnevmatikakis and Giovannucci, 2017*). Event times (Events) were extracted from the deconvolved signals. Signal shown is depicted in the boxed region in D. (**D**) Spatial footprints of 15 Purkinje cell dendrite arbors (A) and their corresponding calcium transients (C_raw), event raster across all (168) extracted signals, the x, y, z accelerometer channels, as well as the mean raw signal (C_raw mean) and sum of all events (Event sum). (**E**) Purkinje cell dendrite transients were aligned to acceleration onset in the red shaded area in D. In this example a reflexive movement (twitch) was triggered by a loud clap. (**F**) Spontaneous behaviors monitored with a webcam can be associated with distinct signatures in the accelerometer data such as during rearing. In this recording a subset of Purkinje cells showed a significant response during animal rearing with transient elevations associated with the lifting and falling phase of the mouse (average of 4 rearings, mean ± SEM, N = 1 mouse). The dashed lines indicate the time points at which the webcam images were captured.

The online version of this article includes the following video, source data, and figure supplement(s) for figure 2:

**Source data 1.** Cerebellar imaging with NINscope.
**Figure supplement 1.** Excitation LED light power as a function of current supplied before and after the GRIN objective and relay lenses.
**Figure supplement 1—source data 1.** Excitation LED light power as a function of current supplied measured before and after the GRIN objective and relay lenses.
**Figure 2—video 1.** Cerebellar imaging with NINscope.
https://elifesciences.org/articles/49987#fig2video1

dendrites, which each span up to ~150 μm in width and have a thickness of ~4–6 μm (*Nedelescu and Abdelhack, 2013*), make them amenable for miniscope imaging (*Ghosh et al., 2011*). We were thus able to extract parasagittally aligned spatial footprints using CNMF-E (see Materials and methods for specific parameter settings).

For all recordings minimal light power (~120–240 μW before, 30–140 μW after entering the GRIN objective, well within the linear range of the excitation LED driver, *Figure 2—figure supplement 1*) was required to obtain high quality signal-to-noise recordings using a GRIN objective mounted on the brain surface (*Figure 2D*, *Figure 2—video 1*). No signal bleaching or photodamage was observed for the duration of our imaging sessions (typically 5.000–20.000 frames, that is, 2–11 min per session repeated with brief intermissions at least 10 times). The x, y and z accelerometer channels were used to detect movement onsets (*Figure 2E*) and discern specific behaviors such as rearing (*Figure 2F*).

## Dual-region imaging with NINscope

An incentive to build lighter and smaller miniscopes is the ability to record from two regions concurrently in unrestrained mice without disrupting spontaneous behavior. For example, one could obtain concurrent cellular resolution recordings of cerebellum and cerebral cortex. There is ample anatomical evidence for cerebello-thalamo-cerebral loops (*Hoover and Strick, 1999*; *Akkal et al., 2007*; *Bostan et al., 2013*) and an increasing number of studies suggest that functional interactions within such loops are important for the proper expression of social, cognitive and motor (planning) behaviors (*Badura et al., 2018*; *Stoodley et al., 2017*; *Gao et al., 2018*). The ability to record from both cerebellum and cortex at once in unrestrained mice opens up the possibility to study how these interactions play out during natural spontaneous behaviors. Moreover, bihemispheric recordings of the hippocampus (*Gonzalez et al., 2019*), neocortex, or cerebellum can reveal the stability, redundancy and lateralization of neural representations during learning and behavioral control.

Given the reduced footprint and weight of NINscope, we were able to mount two miniscopes on a single mouse (*Figure 3A*). We chose to record from cerebellum and cortex to determine to what extent the acceleration of movement correlated with neural activity in these two regions, something that could be directly addressed by using NINscope given its built-in accelerometer.

With two miniscopes mice engaged in cage exploration, grooming, eating, jumping (*Figure 3—video 1*) and rearing as observed in mice with a single miniscope. We quantified the number of rearings as well as their duration in mice wearing one or two miniscopes (*Figure 3B*). Both the number of rearings and their duration were unaffected in mice wearing two miniscopes (number of rearings one scope = 1.47 ± 1.07 rearings/min. mean ± SD; number of rearings two scopes: 1.36 ± 0.75 rearings/min. mean ± SD, p=0.7864, n.s.; rearing duration one scope = 2.21 ± 1.11 s mean ± SD; rearing duration two scopes = 1.84 ± 0.77 s mean ± SD, p=0.4244, n.s.; N = 4 mice, Kolmogorov-Smirnov test). In addition to mounting one scope above cerebellum and one above cortex, we tested the ability of mice to wear dual scopes in two alternate configurations (*Figure 3—figure supplement 1*). One configuration with miniscopes mounted above two hemispheres of visual cortex, which also opens up the possibility to record from the underlying hippocampal CA1 region (number of rearings: 1.5 ± 0.95, mean ± SD; rearing duration: 2.4 ± 1.15, mean ± SD, N = 1 mouse). Another configuration with two miniscopes mounted above the (Crus I) cerebellar hemispheres (number of rearings: 1.8 ± 0.6, mean ± SD; rearing duration: 1.84 ± 0.82, mean ± SD, N = 1 mouse). For these two additional configurations the number of rearings and their duration were comparable to the single scope or cerebellum-cortex configurations (*Figure 3—figure supplement 1*). A nested ANOVA to compare across all miniscope (single and dual) configurations showed that neither rearing duration ($F_{(3,6)}$:1.859, p=0.23) nor rearing frequency ($F_{(3,6)}$:0.221, p=0.88) were significantly different. A post-hoc Tukey's test for multiple comparisons confirmed that none of the interactions among the different configurations were significant (*Figure 3—source data 1*).

Our miniscope design allows recordings from two regions in mice with an inter-baseplate distance of ~8 mm and miniscopes placed at an angle of 15–20° in the case of concurrent recordings from cerebellum and cortex (or alternatively dorsal striatum, *Figure 3C,D*), and ~6–7 mm at an angle of 45–50° when placed side-by-side for bihemispheric recordings from cerebellum or cortex. These values were obtained using accurate three-dimensional models of a C57BL/6 mouse skull and two NINscopes with baseplates using CAD modeling software. Thus, based on our findings, NINscope permits multi-configuration dual site recordings in unrestrained mice (*Figure 3—figure supplement 2*).

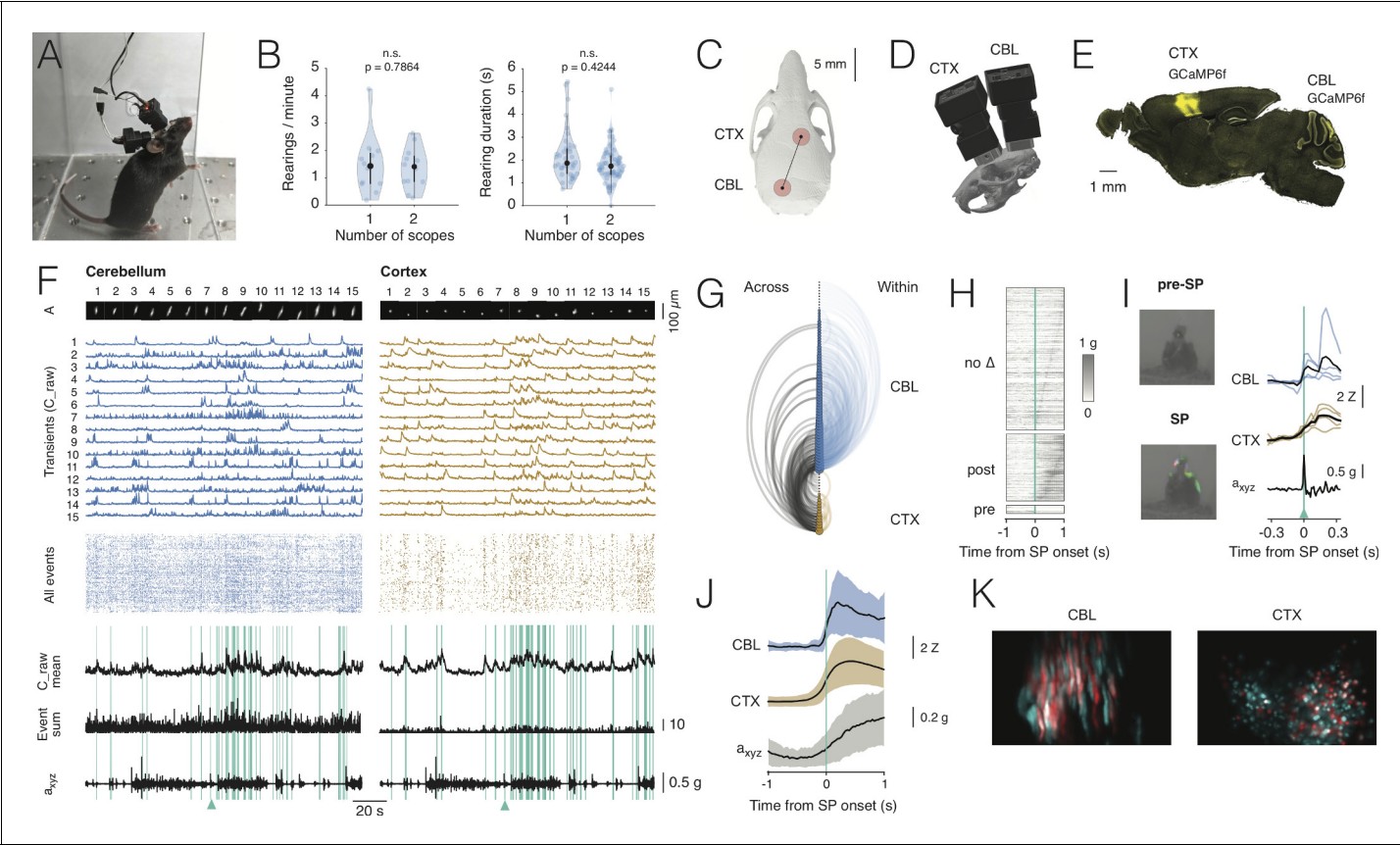

**Figure 3.** Dual-region imaging with NINscope. (**A**) A mouse with two NINscopes mounted over cerebellum and cortex. (**B**) Behavior was unimpaired as quantified by counting the number of rearings and their duration in mice wearing single or dual miniscopes (rearings/minute, p=0.7864, n.s.; rearing duration, p=0.4244, n.s.; N = 4 mice). (**C**) Mouse skull with red circles indicating the recording configuration (craniotomy positions). For other possible configurations see *Figure 3—figure supplement 1* and *Figure 3—figure supplement 2*. (**D**) CAD rendering showing the rostro-caudal placement of two NINscopes to image from cerebellum an cortex concurrently (~8 mm inter-baseplate distance at angles of 15–20°). (**E**) GCaMP6f was transduced selectively in cerebellar Purkinje cells (lobule VI or simplex lobule) and in neurons of motor cortex. (**F**) Dual site recordings from cerebellar lobule VI and motor cortex showing responses of segmented neurons (spatial footprints, A) in each region with the Z-mean scored signal, number of co-active Purkinje cell dendrites and compound acceleration signal ($a_{xyz}$, $\sqrt{(x^2+y^2+z^2)}$). Cyan lines represent epochs where synchronous patterns (SPs) were found across cerebellum and cortex. (**G**) Combined arc plots for this dataset visualizing intra-cerebellar, intra-cerebral (within) and cerebello-cerebral (across) SPs. Node radii scale by the number of cells that a node connects to. In this example cerebellar neurons with high within SPs also displayed significant SPs across regions. (**H**) SPs were used to trigger the compound acceleration signal. Behavioral acceleration could be assigned to four categories consisting of no change (no Δ, 64%, sorted by peak response), behavioral acceleration post-SP (31%), pre-SP (4%), or around-SP (1%, not shown). (**I**) Across-regions SPs are associated with significant deviations from baseline in the accelerometer compound signal. Responsive cells are shown in cerebellum (lobule VI) and cortex, triggered off of the SP. A mouse resting prior to an SP that made a (left, upward) movement around SP onset. Animal movement visualized with optic flow is color-coded. (**J**) Population averaged responses triggered around detected SPs reveal responses in cerebellum and cortex during accelerometer upslope. (**K**) Example showing neurons in the cerebellum and cortex that participated in an SP (red) and cells that did not (cyan).

The online version of this article includes the following video, source data, and figure supplement(s) for figure 3:

**Source data 1.** Dual-region imaging with NINscope.

**Figure supplement 1.** Tested dual NINscope configurations.

**Figure supplement 1—source data 1.** Data pertaining to rearing behavior with various configurations of NINscope.

**Figure supplement 2.** Possible dual NINscope configurations.

**Figure supplement 3.** Bead imaging to assess motion artifacts with dual miniscopes.

**Figure supplement 3—source data 1.** Data related to x and y shifts for beads injected in cerebellum and cortex.

**Figure 3—video 1.** A jumping mouse wearing two NINscopes.

https://elifesciences.org/articles/49987#fig3video1

**Figure 3—video 2.** Motion correction and segmentation in mice wearing dual miniscopes.

https://elifesciences.org/articles/49987#fig3video2

**Figure 3—video 3.** Bead imaging with dual NINscopes to assess motion artifacts.

*Figure 3 continued on next page*

*Figure 3 continued*

https://elifesciences.org/articles/49987#fig3video3

**Figure 3—video 4.** Dual-region imaging with NINscope.

https://elifesciences.org/articles/49987#fig3video4

In our specific demonstration of dual site imaging, virus injections were made to selectively transduce GCaMP6f in cerebellar Purkinje cells of cerebellar simplex lobule (GRIN objective center AP:−5.8, ML:2.2 mm) or lobule VI (GRIN objective center AP:−7.4, ML:0.0 mm), as well as globally in neurons of motor cortex (GRIN objective center AP: +1.4, ML: 1.5), covering large parts of the caudal forelimb area and a small part of M2 at the rostral border of the GRIN objective lens (*Figure 3E*).

Brain motion artifacts are common during in vivo imaging in both head-fixed and unrestrained animals and algorithmic approaches have been developed to correct for them (*Pnevmatikakis and Giovannucci, 2017*). Motion artifacts in our data were apparently modest even in mice wearing dual miniscopes (*Figure 3—video 2*), which we further checked by injecting static fluorescent beads into the cerebellum and cortex to monitor bead movement during unrestrained behavior (*Figure 3—figure supplement 3*, *Figure 3—video 3*).

Signals could be extracted from hundreds of cells in the cerebellum and cortex (cerebellum: 141 ± 53, range 62–200, cortex: 201 ± 90, range 89–361, mean ± SD, N = 4 mice, *Figure 3F*). Calcium transients from Purkinje cell dendrites expressing GCaMP6f displayed faster decay kinetics than cells in cortex expressing the same calcium sensor protein ($t_{1/2}$ cerebellum: 0.217 ± 0.12 s, 15784 transients; $t_{1/2}$ cortex: 0.488 ± 0.34 s, 9424 transients, mean ± SD, Hedge's G = 1.19), likely reflecting differences in endogenous calcium buffering (*Celio, 1990*; *Baimbridge et al., 1992*). Event rates measured from Purkinje cell dendrites corresponded to the underlying rate of climbing fiber input (0.62 ± 0.39 Hz, mean± SD, 1629 cells, N = 4 mice), while cortical neurons fired events at a lower rate (0.22 ± 0.28 Hz, mean ± SD, 2212 cells, N = 4 mice).

In all recordings we observed distinct periods of increased activity across regions (*Figure 3—video 4*) around periods of animal movement (*Figure 3F*), as gauged from the compound acceleration signal and inspection of the raw data. To quantify correlations across cerebellum (lobule VI, N = 2, simplex lobule, N = 2; N = 4 mice) and cortex, calcium transient onset times, extracted from deconvolved transients, were convolved with an Epanechnikov kernel and summed over all cells. This resulted in a kernel sum, in which global synchronous patterns (SPs) were defined as instances where the kernel sum exceeded mean+2σ and where these patterns occurred at least five times (cyan lines in *Figure 3F*). A visualization of the functional connectivity of cells is shown in the combined arc plots of *Figure 3G*, where node size scales with the number of SPs within cerebellum (lobule VI in this example), within cortex, or across the two regions. Both cerebellar Purkinje cells and cortical neurons displayed SPs within and across regions, with Purkinje cells having significantly more within-SPs than cortical neurons (fraction of total lobule VI: 66.4 ± 18.9%, simplex lobule: 73.3 ± 27.4%, cortex: 35.7 ± 17%, mean ± SD, CBL vs CTX p=2.7801e-17, Kolmogorov-Smirnov test). Significantly more Purkinje cells participated in across-SPs than cortical neurons (fraction of total lobule VI: 42.3 ± 29.6%; fraction simplex lobule: 47.4 ± 31.6%; cortex: 26.7 ± 18.5%, mean ± SD; CBL vs CTX p=1.8409e-15, Kolmogorov-Smirnov test).

We next assessed whether correlated activity in cerebellum and cortex were associated with behavioral acceleration. To determine if it preceded or followed across-region-SPs, the compound acceleration signal was triggered off of across-SPs. We could assign behavioral acceleration to four categories (*Figure 3H*), consisting of no significant change (64%, 206/321 across-SPs), behavioral acceleration post-SP (31%, 99/321), pre-SP (4%, 13/321) or around-SP (1%, 3/321). Thus, the majority (86%, 99/115) of SPs associated with significant acceleration events following the across-SP. We further confirmed that SPs are associated with acceleration events by comparing the fraction of randomly triggered versus SP-triggered acceleration exceeding mean+2σ of baseline to show that the fraction of SP-triggered acceleration events could not have arisen by chance (random vs SP-triggered, p=0.0017, Kolmogorov-Smirnov test).

As expected, calcium transients in cerebellar Purkinje cells and cortical neurons occurred around behavioral acceleration (*Figure 3I,J*). In the example shown (triggered off an across-SP indicated by

the cyan triangles in *Figure 3F and I*), a large deflection occurred in the compound accelerometer signal around an SP onset. This movement was also seen when inspecting the concurrent webcam image frames and visualized here by optic flow, with colors representing direction of movement before (top) and around the SP onset (bottom).

Aligning recordings (N = 4 animals, nine recordings) to post-SP onsets revealed a clear upslope of the accelerometer signal coinciding with calcium increases in both cerebellar Purkinje cells and cortical neurons (*Figure 3J,K*). Although electrical recordings provide more accurate latency estimates than calcium imaging, we examined the relative timing of the calcium responses in cerebellum and cortex relative to movement acceleration. Latencies to peak response of the calcium transients in cerebellum, cortex and the compound acceleration signal revealed that the maximum acceleration came after the peak response in both cerebellum and cortex (CBL: 444 ± 280 ms, CTX: 505 ± 260 ms, $a_{xyz}$: 700 ± 269 ms, Kruskal-Wallis test p=1.6553e-08). A post-hoc test using Sheffe's S revealed that the mean ranks for the calcium transient peak latencies were both significantly different relative to the delayed peak latency of the accelerometer signal, while this did not apply when comparing the mean ranks of cerebellar and cortical calcium transients. Thus, based on our preliminary data recorded with NINscope, we show that coordinated cerebello-cerebral activity generally precedes peak acceleration of a movement.

## Multi-site optogenetic stimulation

So far, miniscope optogenetics has focused on stimulating within the same field-of-view (*Stamatakis et al., 2018*). Stimulating within the same field-of-view sets constraints on the miniscope design necessitating additional optics, which adds weight and increases the footprint of a miniscope. It also requires the use of an opsin that can be spectrally separated from the activity indicator. Moreover, it is often desirable to stimulate at a site distal to the miniscope, for example to activate indirect projection pathways. By directly driving the LED from the miniscope, the amount of cabling can be reduced, while stimulus onsets can be directly logged to disk together with the image frames and accelerometer data to ease post-hoc analysis. In addition, multiple probes can be used to stimulate in more than one location within the same animal. Using the integrated optogenetic LED driver of NINscope we demonstrate its use by stimulating cerebellar Purkinje cells in transgenic *Pcp2*-Cre Jdhu x Ai32(RCL-ChR2(H134R)/EYFP) mice, while performing calcium imaging from neurons in motor cortex transduced with GCaMP6f (*Figure 4A–B*, GRIN objective center AP: +1.4, ML: 1.5). We chose four cerebellar locations to place the blue (470 nm) optogenetic LED probes: Crus II ipsi- and contralateral to the imaging site on the right hemisphere, contralateral simplex lobule and lobule VI in medial cerebellar vermis. LED probe connector pins (*Figure 4C*) allowed switching the miniscope LED driver connection from one implant site to the next (*Figure 4D*).

Stimulation of each of the cerebellar sites (50 ms, 22 mA current, 2.3 mW light power) resulted in an increased number of activated neurons in the motor cortex of the right hemisphere (*Figure 4E–G*, *Figure 4—video 1*). Recurring cerebellar stimulation (0.3 Hz, 4–5 times) could induce ramp-like activity as seen when averaging responses across cells in motor cortex (*Figure 4E*). The miniscope accelerometer data registered movement that occurred with a delay after stimulus offset (crus II ipsi: 76 ± 23 ms; crus II contra: 80 ± 30 ms; lobule VI: 79 ± 34 ms; simplex lobule: 73 ± 32 ms, mean ± SD, N = 2 mice) in line with earlier observations that disinhibition of Purkinje cells and associated rebound excitation of neurons in the cerebellar nuclei can elicit delayed motor reflexes (*Hoebeek et al., 2010*; *Witter et al., 2013*). A significant number of cells met the criterion of displaying calcium increased exceeding mean+2σ of the pre-stimulus baseline (referred to as responders; see also *Figure 4F*). Across all recordings and sites of stimulation the number of responders was roughly half of all segmented cells in motor cortex (49.9 ± 14.2%, mean ± SD, 2227/4614 cells, N = 2 mice). Mean amplitudes of Z-scored calcium transients measured in neurons of motor cortex varied depending on the region stimulated (crus II ipsi: 2.1 ± 1.3 Z; crus II contra: 2.9 ± 1.5 Z; lobule VI: 2.9 ± 1.6 Z; simplex lobule: 2.0 ± 1.3 Z; Kruskal-Wallis test p=1.1239e-70 and post-hoc multiple comparisons using Sheffe's S revealed significantly different mean ranks for stimulation of contralateral lobule VI and crus II relative to the other groups) (*Figure 4G*), but all displayed comparable onset times (crus II ipsi: 218 ± 120 ms; crus II contra: 221 ± 118 ms; lobule VI: 233 ± 123 ms; simplex lobule: 212 ± 130 ms, mean ± SD; Kruskal-Wallis test, p=0.1, n.s) relative to stimulus offset. Stimulation of Purkinje cells over both cerebellar hemispheres evoked opposing head movements as determined from our accelerometer data with leftward movements when stimulating over left and rightward

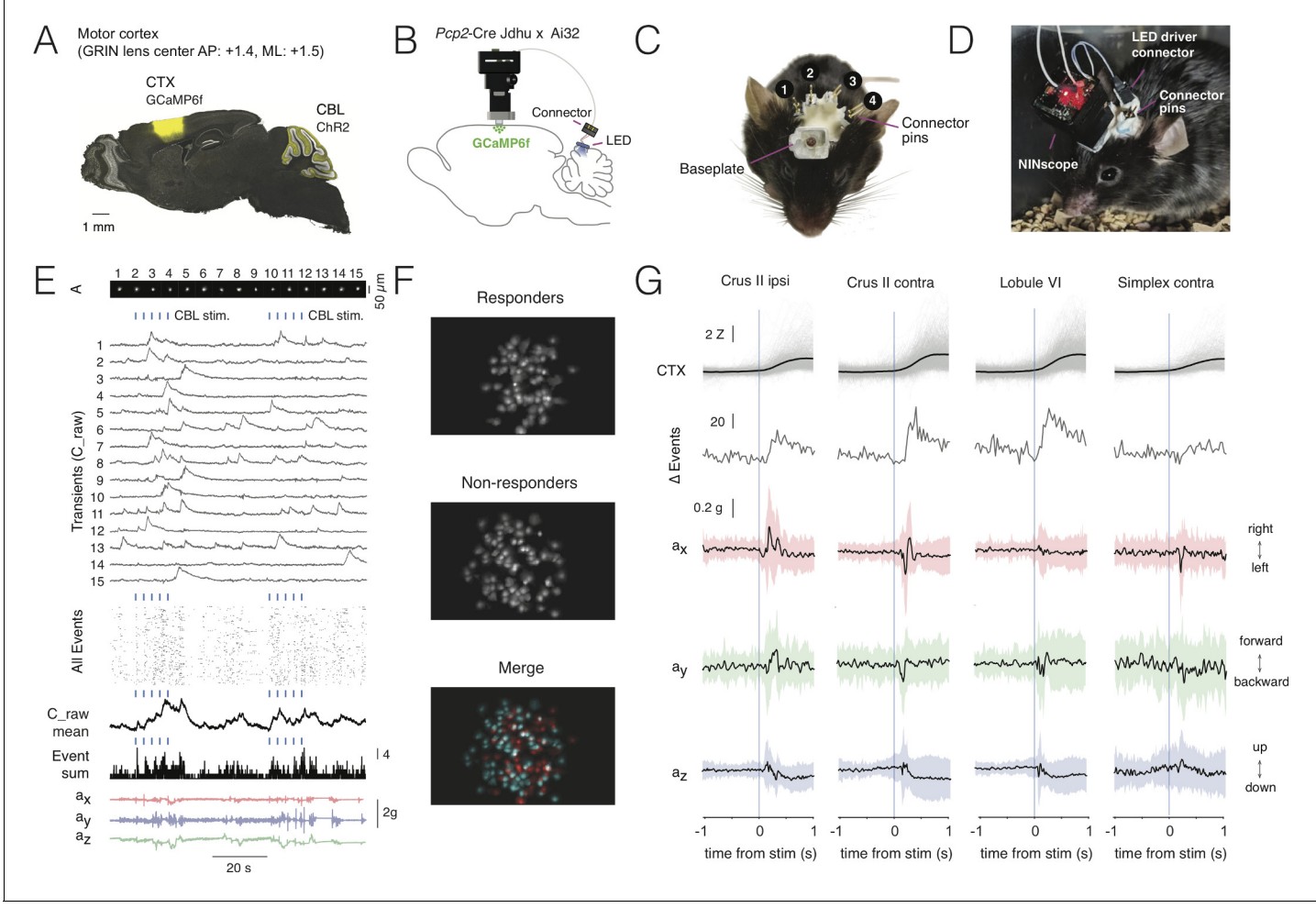

**Figure 4.** Multi-site optogenetic stimulation with NINscope. (**A**) *Pcp2*-Cre Jdhu mice were crossed with Ai32 mice to obtain selective expression of ChR2(H134R) in cerebellar Purkinje cells. Neurons of the motor cortex were transduced in these mice with GCaMP6f. (**B**) Experimental configuration to combine optogenetic stimulation of cerebellum with imaging in cortex. (**C**) Mouse with a baseplate above the cortex and four LED probes mounted above the cerebellum (1: Crus II ipsi, 2: lobule VI, 3: Crus II contra, 4: Simplex lobule). The connector pins were used to connect the NINscope LED driver to each of the four probes. (**D**) Mouse with a NINscope and connection to one of the four stimulation sites. (**E**) Optogenetic stimulation of Purkinje cells (50 ms, 22 mA, 2.3 mW) evoked clearly discernible increases in both mean response (C_raw mean) and number of co-active cells (spatial footprints, A) in motor cortex (All Events, Event Sum). Repeated stimulation induced ramp-like activity in cortex. (**F**) In this example a large fraction of cells responded (responders) to optogenetic stimulation of the contralateral cerebellar hemisphere (Crus II). Color merge shows spatial localization of responders (red) and non-responders (cyan). Responsive cells were selected using the criterion that the post-stimulus signal had to exceed mean+2σ of the pre-stimulus baseline. (**G**) Calcium transients (gray: individual transients, black: mean) and change of summed events (Δ Events) triggered to stimulus onset at four different locations over the cerebellar surface and corresponding x (red), y (green) and z (blue) channel accelerometer data. Stimulation of the cerebellar hemispheres reveals lateralization of the behavioral response with stimulation on the left or right eliciting leftward and rightward head movements, respectively. Evoked behavioral reflexes generally commenced prior to calcium transient onsets in the cerebral cortex. The online version of this article includes the following video, source data, and figure supplement(s) for figure 4:

**Source data 1.** Multi-site optogenetic stimulation with NINscope.

**Figure supplement 1.** Light stimulation in absence of ChR2 neither evokes cerebral cortical nor behavioral responses.

**Figure supplement 1—source data 1.** Control experiment data in which optogenetic stimulation and imaging was performed in a wildtype mouse lacking ChR2(H134R).

**Figure 4—video 1.** Combining remote optogenetic stimulation with cortical imaging using NINscope.

https://elifesciences.org/articles/49987#fig4video1

movements when stimulating over right crus II (*Figure 4G*). These data are in line with the findings that crus I and II in rodents do not only receive inputs related to orofacial and whisking behavior (*Ju et al., 2019*; *Romano et al., 2018*), but also head and neck information (*Quy et al., 2011*; *Huang et al., 2013*), and that activation of the cerebellar hemispheres in humans is associated with head movements (*Prudente et al., 2015*). Lateral stimulation of left simplex lobule evoked more modest lateral movements as compared to crus II stimulation, whereas they were mostly absent when stimulating over medial vermis lobule VI where forward/backward movements were more pronounced. The behavioral reflexes registered with the accelerometer upon Purkinje cell stimulation did not appear directly correlated with cerebral cortical activation, suggesting other, downstream targets, underlying these reflexes. We did not observe an increase in the number of activated neurons in motor cortex upon stimulation with the same intensity and duration in a wildtype mouse lacking ChR2, nor were such stimulations associated with stimulus-triggered deflections of the accelerometer (*Figure 4—figure supplement 1*).

## Deep brain imaging and behavioral parsing

The striatum is a subcortical structure that is inaccessible to imaging without lowering a GRIN relay lens to the site of interest (*Figure 5A*). Due to tissue damage along and below the lens track, longer recovery times are required before imaging can commence (*Bocarsly et al., 2015*). A significant amount of light is lost through a combination of two GRIN lenses along the optical path (GRIN relay and GRIN objective lens), thereby rendering these experiments more challenging than imaging from cerebellar Purkinje cell dendrites or superficial layers of cerebral cortex. In order to validate NINscope to study the striatum in unrestrained animals and in particular to prove its effectiveness for deep-brain imaging, we revisited previous work that has proposed a role of the dorsal striatum (DS) in contraversive movement initiation and action encoding (*Klaus et al., 2017*; *Cui et al., 2013*).

Using a viral vector with the human synapsin promoter, we transduced all striatal neurons with GCaMP6s or GCaMP6f in a caudal, dorsal part of the right striatum (*Figure 5B*). Directly following viral transduction, a 600 µm diameter GRIN relay lens was implanted above the region of interest. NINscope was mounted on a baseplate that had the GRIN lens objective glued in place. The whole assembly was lowered to just above the GRIN relay lens to bring cells into focus. In DS, we extracted signals from up to 84 cells (62 ± 16.70, mean ± SD, range 38–84; *Figure 5C*), which, based on their calcium transients, had a rate of ~1 Hz (1.106 ± 0.93 Hz, mean ± SD, n = 308 cells, N = 5 mice) (*Figure 5D*). Despite using relatively low light power (~300 µW after the objective and before the GRIN relay lens), we obtained good signal-to-noise recordings. Using both the NINscope tracking LED and the accelerometer data, we found epochs where mice made both spontaneous body and head turns (*Figure 5E*, *Figure 5—video 1*). Such turns were associated with up- or downward deflections in the x channel of our accelerometer, reflecting left or right-turning movements, respectively (*Figure 5C,D*). During left turns (contralateral to the imaging location in DS), a majority of neurons in the right DS (85%, n = 308 cells, N = 5 mice) had peak responses after action initiation and 20% displayed significantly elevated responses for the duration of action execution (paired t-test baseline vs action execution activity, p<0.05) (*Figure 5G,H*). The largest of these began after movement initiation, suggesting a predominant association with action execution rather than preparation (latency onset, 310 ± 45 ms, mean ± SD). None of the cells we recorded from responded during forward-backward movements or movements ipsilateral to the site of recording (*Figure 5G,H*), confirming lateralization of movement signals in the striatum. When signals were averaged one second pre- and post-movement initiation (*Figure 5I,J*), repeated measures ANOVA revealed a significant main effect ($F_{(1.18,\ 4.7)}$=16.48, p=0.01) of action execution. Post-hoc Bonferroni correction for multiple comparisons demonstrated that the effect occurred exclusively during epochs of contralateral (t = 9.45, df = 4, p=0.001), but not ipsilateral movement initiation (t = 3.07, df = 4, n.s.).

## Combining deep-brain imaging with optogenetic stimulation

To show the utility of NINscope for combining optogenetic stimulation with deep-brain imaging we studied the impact of cortical inputs on neuronal activity in the DS of unrestrained animals. The opsin ChrimsonR was transduced in either orbitofrontal cortex (OFC) or secondary motor cortex (M2) and an LED (with 645 nm peak emission) was placed above the cortex (*Figure 6A*). The OFC and M2 can differentially regulate the activity of neurons in specific DS regions in vitro (*Corbit et al., 2019*), with

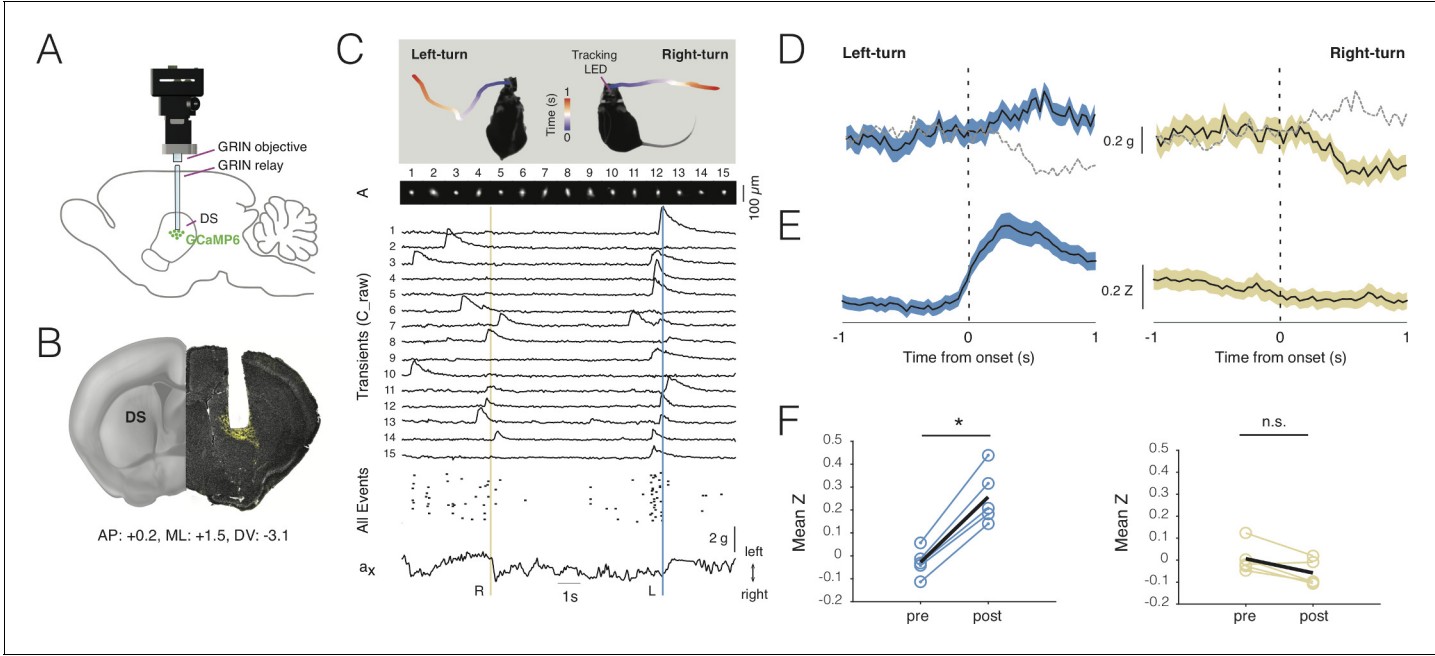

**Figure 5.** Deep-brain imaging and behavioral parsing with NINscope. (**A**) Schematic of the NINscope configuration, which combines a GRIN objective with a GRIN relay lens (600 μm) to image from the dorsal striatum (DS) of the right hemisphere. (**B**) Coronal section showing the GRIN relay lens track and neurons in right DS expressing GCaMP6f (yellow). (**C**) Left- and right-turns were quantified by combining video observations or tracking LED with analysis of G-sensor data. Shown on top are examples of a mouse making a left and right turn in an open-field arena and the path obtained from the NINscope tracking LED where time progression over a duration of one second is color-coded. Below this the spatial components (A) and transients (C_raw) of 15 neurons extracted with CNMF-E are shown, the onset times of all events extracted from C as well as the x channel of the accelerometer. The vertical bars indicate that the animal first turned right (yellow) and then left (blue), with activity modulation in the right DS coinciding with contraversive movements. (**D**) Accelerometer data showing mean left and right acceleration of the x channel around movement onset (mean ± SEM). Dashed gray lines represent contraversive acceleration. (**E**) Mean calcium transient responses one second before and after movement onset (N = 5 mice) reveal a clear modulation of activity during left turns (mean ± SEM) when imaging right DS. (**F**) Quantification of calcium transient responses before and after movement onset for left and right turns, respectively. Right DS only displayed a significant calcium-transient increase for left turns (p<0.05).

The online version of this article includes the following video and source data for figure 5:

**Source data 1.** Deep-brain imaging and behavioral parsing with NINscope.

**Figure 5—video 1.** Deep imaging with NINscope.

https://elifesciences.org/articles/49987#fig5video1

the impact of OFC on DS being stronger than that of M2. We sought to confirm these findings in vivo. We first assessed the direct terminal fields of OFC and M2 to DS and mapped this onto a representative brain atlas image (*Figure 6B*). M2 input to DS was more diffuse than the projections of OFC, consistent with previous findings (*Hintiryan et al., 2016*; *Corbit et al., 2019*; *Hunnicutt et al., 2016*). During imaging sessions of DS transduced with GCaMP6f (light power used: 170 μW after the relay GRIN lens, *Figure 2—figure supplement 1*), animals were able to freely explore an open-field arena while OFC or M2 were optogenetically stimulated (10 s, 20 Hz, 5 ms pulse width, 3.4 mW LED). During such OFC and M2 stimulation the responses of DS neurons could be divided into three distinct types, including those that were decreased, increased, or unchanged relative to baseline activity (paired t-test baseline vs stimulus evoked activity, p<0.05, N = 4 mice) (*Figure 6C*). The firing frequency of the calcium transients during stimulation differed between these clusters (decreased: 0.23 ± 0.25 Hz, mean ± SD, n = 33 cells, N = 4 mice, unchanged: 0.77 ± 0.68 Hz, mean ± SD, n = 199 cells, N = 4 mice, increased: 1.63 ± 1.23 Hz, mean ± SD, n = 22 cells, N = 4 mice).

Stimulation was repeated for 10 trials and responses were averaged over trials (*Figure 6D*). Modulation of activity in subpopulations of DS neurons during OFC stimulation was observed: 20% of the neurons (26/133) displayed a significant decrease in activity, 69% displayed no change (93/133) and 11% (15/133) were increased. M2 stimulation had a comparable, but weaker impact on activity of

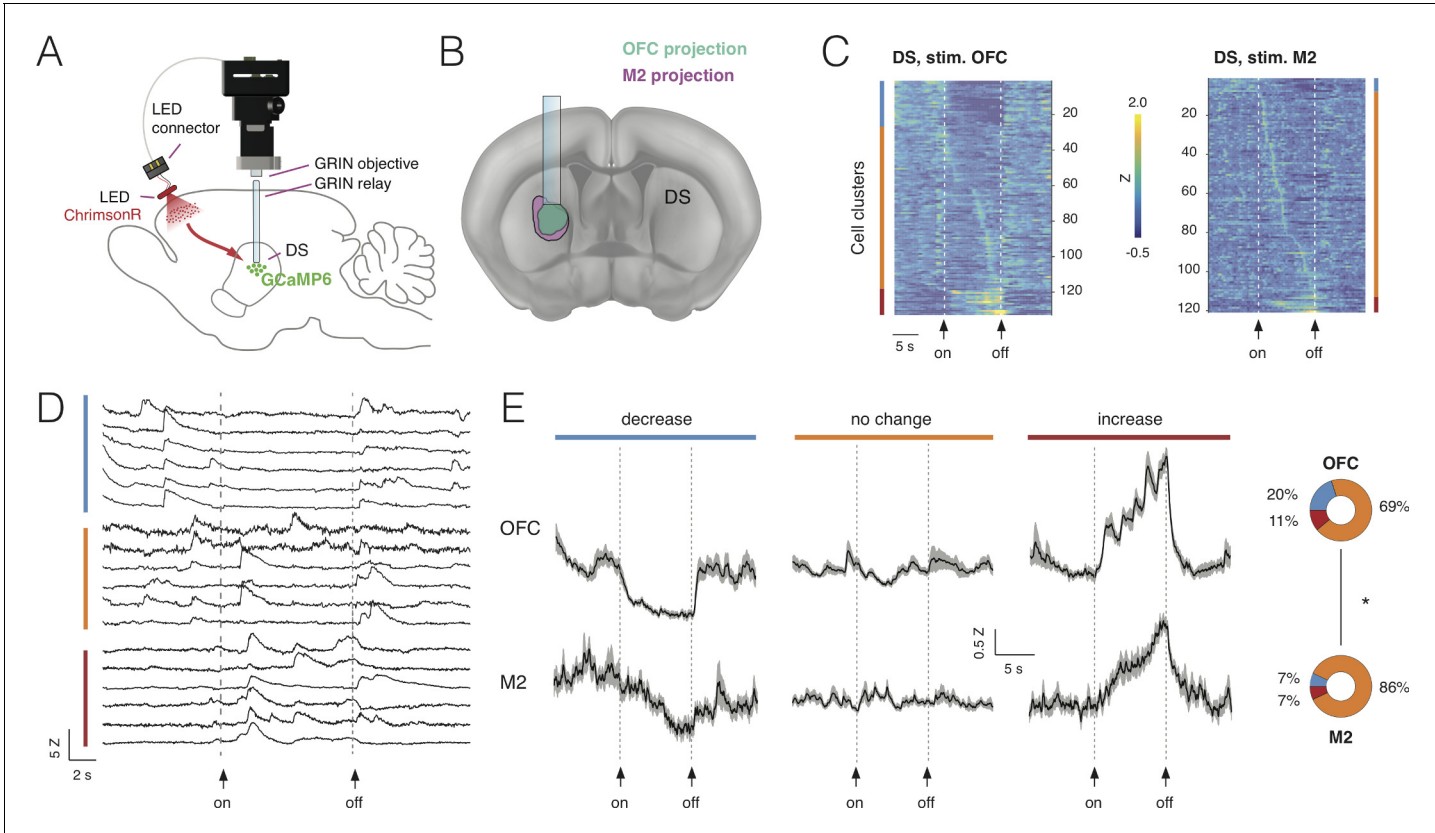

**Figure 6.** Combining deep-brain imaging with optogenetic stimulation using NINscope. (**A**) Schematic showing placement of the NINscope with GRIN objective and GRIN relay lens to record from dorsal striatum (DS) as well as location of the optogenetic LED probe driven by the integrated LED driver. Viral vectors were injected either in orbitofrontal cortex (OFC) or secondary motor cortex (M2) to transduce neurons with ChrimsonR, or in DS to transduce neurons with GCaMP6f for calcium imaging. (**B**) Terminal fields of OFC and M2 mapped onto an Allen Brain Atlas template show their overlap in DS underneath the GRIN relay lens. (**C**) Responses for each neuron averaged over 10 trials for 254 cells in four mice. Different types of responses are found in DS when either OFC or M2 are optogenetically stimulated. Neurons exhibited decreases of activity (blue cluster), no apparent change (orange cluster), or increased responses (red cluster). (**D**) Z-scored calcium transients during OFC (left, N = 2 mice) and M2 (right, N = 2 mice) stimulation (10 s pulse, 20 Hz). For each neuron, 10 trials were averaged. (**E**) Responses averaged over all DS neurons for all 10 trials revealed similar types of modulation during stimulation of OFC and M2 (mean ± SEM). The circular insets denote the fraction of cells that showed suppression, no change, or increased responses.

The online version of this article includes the following source data for figure 6:

**Source data 1.** Combining deep-brain imaging with optogenetic stimulation using NINscope.

subpopulations of DS neurons with 7% of neurons (7/107) showing a decrease, 86% (93/107) displaying no change and 7% (7/107) of cells showed an activity increase. Although the fraction of responsive DS neurons (i.e. those showing a decrease or increase) during cortical stimulation significantly differed between OFC (31%) and M2 (14%) (chi-squared(1)=13.85, p<0.001), the average response for each category of DS neurons during stimulation was comparable (*Figure 6E*). Decreased neurons showed a gradual reduction in activity for the duration of stimulation, whereas neurons responding to stimulation with increased activity showed a progressive increase as long as the stimulation was provided, suggesting that input modulation of activity in DS neurons scaled with stimulation duration.

## Discussion

### NINscope

We have demonstrated the applicability of NINscope to perform dual site recordings in mice, and combined superficial and deep brain imaging with optogenetic stimulation, while parsing movements through an integrated accelerometer.

Our miniscope adds functionality while retaining a small footprint, making it a valuable contribution to the expanding tool chest of open-source miniscopes. NINscope acquisition and control software is platform-independent and can be used on all major operating systems using different hardware configurations including laptops that are within reach of most users.

A number of design choices had to be made during NINscope prototyping. We 3D printed the microscope housing to keep the design light, but eschewed the use of an electrowetting lens, which would have made focusing practical, but the miniscope too heavy and bulky for dual site recordings. The inclusion of a high-resolution LED driver for optogenetic stimulation together with the use of a custom LED probe provided the ability to directly drive projections at their site of origin instead of their terminations within the imaging field-of-view, which is a feature that has not been integrated into a miniscope before. In addition, we present the first miniscope with an integrated accelerometer that allows parsing of behavior during imaging. Finally, we have included a second excitation LED driver on the interface PCB that would permit two-color fluorescence excitation in future designs.

Given that our building plans are open-source, the core functionality of our scope can be expanded depending on the specific research questions that need to be addressed, which ultimately dictate the size, weight and functionality constraints.

### Dual region imaging and multi-site optogenetics

To our knowledge we present the first concurrent cellular resolution recordings of cerebellum and cerebral cortex in unrestrained mice. We find that synchronous patterns of activity across these regions are common across multiple behavioral states and correlate with movement acceleration. Concurrent imaging of cerebellum and cerebral cortex could help to address how activity across the two areas correlates during the planning, learning and execution stages of a movement (*Guo et al., 2015*; *Kawai et al., 2015*; *Sauerbrei et al., 2020*; *Gao et al., 2018*). Using NINscope we also performed a crude assessment of functional connectivity between cerebellum and cerebral cortex by optogenetically stimulating Purkinje cells at multiple cerebellar locations in the same animal. Such stimuli evoked reflexive movements and subsequent activation of neurons in the cerebral cortex with delays similar to those that we have reported previously (*Witter et al., 2013*). NINscope can thus facilitate studies that require optogenetic stimlation of multiple distinct sites with read-out of activty in one or two remote regions. Dual-region imaging using NINscope is not limited to recordings from cerebellum and cortex or subcortical structures like the rostral striatum. We show that other configurations are possible including bihemispheric recordings of cortex, hippocampus and cerebellum. Such types of recordings could reveal to what extent neural activity is lateralized during behavior and learning and could also track the stability of multi-region neural signals over time (*Gonzalez et al., 2019*). With integrated control over optogenetics and read-out of acceleration signals NINscope is posited to enable future closed-loop control of activity triggered off of either behavioral state or activity levels as measured in one or two regions.

### Deep brain imaging

The PYTHON480 CMOS sensor incorporated in NINscope enables recording signals from the dorsal striatum (DS) with good signal-to-noise at powers of a few hundred µW before the relay GRIN lens confirming the applicability of our miniscope to image from deep-brain structures. By analyzing the data collected both via video camera and the built-in accelerometer in combination with imaging from DS we could confirm previous findings that identified the striatum as a structure representing action space (*Klaus et al., 2017*; *Cui et al., 2013*; *Barbera et al., 2016*; *Tecuapetla et al., 2014*).

Using NINscope to record from DS in combination with optogenetic stimulation of OFC or M2 at their site of origin, we show that these two input pathways have a differential impact on DS activity as shown previously in brain slices (*Corbit et al., 2019*). Irrespective of the sign of modulation, we

found responses that scaled with the duration of the stimulus, further highlighting the efficacy of NINscope to address questions that require a combined stimulation and imaging approach.

Taken together, NINscope is a versatile open-source miniscope that could fit a niche of users that desire dual region recordings in unrestrained animals, OS interoperability, optogenetic manipulation of areas at a distance from the image site, behavioral parsing using an accelerometer, or a combination of all of the above.

# Materials and methods

**Key resources table**

| Reagent type (species) or resource | Designation | Source or reference | Identifiers | Additional information |
|---|---|---|---|---|
| Genetic reagent (*M. musculus*) | B6.Cg-Tg (Pcp2-cre)3555 Jdhu/J | PMID:15354293 | IMSR Cat# JAX:010536, RRID:IMSR_JAX:010536 | https://www.jax.org/strain/010536 |
| Genetic reagent (*M. musculus*) | Ai32 | PMID: 22446880 | IMSR Cat# JAX:012569, RRID:IMSR_JAX:012569 | https://www.jax.org/strain/012569 |
| Recombinant DNA reagent | AAV1.CAG.flex. GCAMP6f. WPRE.SV40 | Addgene | RRID:Addgene_100835 | https://www.addgene.org/100835/ |
| Recombinant DNA reagent | AAV1.CMV.PI. Cre.rBG | Addgene | RRID:Addgene_105530 | https://www.addgene.org/105530/ |
| Recombinant DNA reagent | AAV.Syn. GCaMP6f. WPRE.SV40 | Addgene | RRID:Addgene_100837 | https://www.addgene.org/100837/ |
| Recombinant DNA reagent | AAV.Syn. GCaMP6f. WPRE.SV40 | Addgene | RRID:Addgene_100843 | https://www.addgene.org/100843/ |
| Other | NINscope | This paper | RRID:SCR_017628 | https://github.com/ninscope |
| Software, algorithm | FIJI | PMID: 22743772 | RRID: SCR_002285 | https://imagej.net/Fiji/Downloads |
| Software, algorithm | MATLAB | MathWorks | RRID: SCR_001622 | https://uk.mathworks.com/products/matlab.html |
| Software, algorithm | Bonsai | PMID: 25904861 | RRID:SCR_017218 | https://bonsai-rx.org/ |
| Software, algorithm | R | R project | RRID:SCR_001905 | http://www.r-project.org/ |
| Software, algorithm | Python | Python | RRID:SCR_008394 | http://www.python.org/ |
| Software, algorithm | SPSS | IBM | RRID:SCR_002865 | http://www-01.ibm.com/software/uk/analytics/spss/ |
| Software, algorithm | Prism | Graphpad | RRID:SCR_002798 | http://www.graphpad.com/ |
| Software, algorithm | Custom Python scripts | This paper | | https://github.com/Romanovg185/sps-continuous-time-data |

## Printed circuit board design

NINscope uses two printed circuit boards (PCBs): a CMOS image sensor and interface PCB each 10 by 10 mm (thickness 0.6 mm, HDI standard) and developed using the open-source and cross-platform electronic design automation suite KiCad (http://kicad-pcb.org). The interface PCB includes the DS90UR913A (Cypress Semiconductor) serializer to connect via an FPD-link III to the DAQ board. The interface PCB is placed on top of the sensor PCB and the two PCBs are connected by wires soldered to the castellated holes at the PCB edges. The sensor PCB contains voltage regulators, a

power sequencer and an oscillator necessary for initializing and operating the image sensor. The voltage regulators are ultra-low noise LDO regulators (NCP163, On Semiconductor) and are specified for use in camera applications. The image sensor needs three power sources that are provided in a sequence, which we achieve by using a LM3880 power sequencer (Texas Instruments). A 66.6667 MHz low power CMOS clock oscillator (Kyocera Electronics) is used to run the image sensor. In addition to the serializer, the interface PCB has an IMU sensor (LSM6DSLTR, STMicroelectronics), 2 LED drivers (single LED for optogenetic stimuli: LM36011; dual LED for one fluorescence excitation LED: LM3643, Texas Instruments) and an I2C I/O expander (FXL6408UMX, ON Semiconductor). The I/O expander controls signals that are not time critical for the image sensor and the tracking/indicator LED on top of the interface PCB. The LED driver and accelerometer are described in more detail below.

## CMOS image sensor

The sensor PCB includes the PYTHON480 (ON Semiconductor) CMOS SVGA image sensor. The PYTHON480 CMOS image sensor is flexible and has many possible configurations. The challenge was to configure it to our demands. Once we figured out how to start the sensor in CMOS instead of LVDS mode we were able to provide the data signals to the general programmable interface (GPIF) of the Cypress USB controller in order to transfer the image data. To optimize power consumption and limit overheating we opted to switch off the Phase-Locked Loop (PLL) to reduce current consumption by 30%. Without the PLL, the CMOS parallel clock output is 4x times lower constraining the amount of pixel data that can be read-out for a given frame rate. We set acquisition rate in the current NINscope to 30 frames per second and the frame size to 752 × 480 pixels to allow read-out of both pixel data and three accelerometer channels from the IMU. Improved thermal dissipation in future iterations of NINscope will make data at higher frame rates up to 120 Hz accessible and allow additional read-out of the gyroscope data from our IMU.

## Data acquisition hardware

The UCLA Miniscope project has the largest open-source miniscope community with a user base of hundreds of users. We made use of the existing data acquisition (DAQ version 3.2) hardware from the UCLA Miniscope project (http://www.miniscope.org) and introduced minor modifications consisting of a 256 kB x 8-bit I2C EEPROM (STMicroelectronics) to hold our larger, modified firmware and connections on the DAQ PCB between GPIO2 to TP4 and SPI signals to GPO0-GPO3. The PYTHON480 CMOS sensor connects via an FPD-Link III and coaxial cable to the Cypress USB controller on the DAQ board. The image sensor has an SPI instead of an I2C interface for sensor configuration. The general purpose outputs (GPO) are used since the FPD-link in the UCLA DAQ supports an I2C and not an SPI bus. This requires routing signals from MOSI, SCKand SS to GPO0, GPO1 and GPO3 allowing full control of the image sensor from the firmware of the USB-controller. The NINscope firmware is based on example firmware derived from the Cypress application note AN75779 (modification date: 30/10/2017). The CMOS image sensor which is registered as a USB imaging device is expanded to function as a composite USB device to allow the addition of a virtual serial port. The virtual serial port is used for communication and configuration of the CMOS sensor and LED drivers (e.g. to set image sensor gain, brightness, black levels, LED settings, optogenetic stimulus parameters) as well as for logging of the accelerometer data.

A low-power (0.65 mA in high-performance mode) 3D accelerometer and gyroscope iNemo inertial module (LSM6DSLTR, STMicroelectronics), set of LED drivers (single LED for optogenetic stimuli: LM36011; dual LED for one fluorescence excitation LED: LM3643, Texas Instruments) and I/O expander (FXL6408, ON Semiconductor) are controlled over the I2C bus. The remaining GPO is used to control pulse generation and precise timing of the optogenetic stimuli by introducing a wired connection between GPO2 and TP4 in the UCLA DAQ (*Figure 1E*).

The IMU accelerometer has a range of + /- 2 g where each bit corresponds to 0.061 mg. A sample rate of 104 Hz is set, which fills a FIFO buffer with x,y,z data that are read out at the end of every image frame and transmitted over the virtual serial connection with the computer.

## LED drivers and LEDs

A single-LED flash driver (LM36011, Texas Instruments) and dual-LED driver (LM3643, Texas Instruments) were used to respectively control the generation of optogenetic pulses in a 470 nm LED probe (LED 150040BS73240, 402 case size, Wurth Electronics) or 630 nm LED probe (APHHS1005-SURCK, Kingbright) and a 470 nm fluorescence excitation LED (Excitation LED LXML-PB02 470 nm, Lumileds). The optogenetic LED can be adjusted in increments of 11.725 mA and the excitation LED in 1.4mA increments. The maximum current of either LED is dependent on the total current consumption of the camera and type/length of the cable used. A 621 nm LED (SML-P11UTT86R, Rohm) was integrated on top of the interface PCB to allow camera-assisted tracking of animal position and for notification of miniscope connectivity.

## Microscope housing

Microscope housing prototypes were designed using Inventor (AutoDesk) with initial prototypes being printed using a Micro Plus Advantage printer (EnvisionTec) using the RCP30 M resin to allow printing of fine detail. The thinnest functional wall of the final prototype had a thickness of 500 µm. The Form 2 (Formlabs) printer was subsequently used in printing microscope housing with black resin (RS-F2-GPBK-04). The microscope consists of three parts, an upper part to hold the PCBs, plano-convex lens and emission filter, a lower part for housing the optics including the LED die, half ball lens, excitation filter and dichroic mirror and a sliding cover to secure the LED and protect the optical filters. The lower part of the microscope has a small protrusion that locks in a notch of the custom metal baseplate (*Figure 1—figure supplement 2*) and is secured by a set screw to minimize microscope movement. After printing and cleaning the housing with isopropyl alcohol and sand-dusting, the in- and outside housing printed with the EnvisionTec printer was airbrushed (Infinity CR Plus, Harder and Steenbeck, Germany) with a thin coat of black paint (H12 Flat Black, Gunze, Japan).

## Optics

Optical designs were modeled using Zemax Optic Studio (*Figure 1—figure supplement 1*). We used custom-diced excitation (ET470/40 × 3.5×3.5 x 0.5 mm, Chroma), dichroic (T495lpxr 3.5 × 5×0.5 mm, Chroma) and emission filters (ET525/50 m, 4 × 4×1 mm, Chroma), a N-BK7 half ball lens (3 mm diameter, 47–269, Edmund Optics) and a plano-convex lens (4.0 mm diameter, 10.0 mm focal length, 45–429, Edmund Optics) to focus onto the CMOS image sensor. The emission filter was bonded to this lens with optical adhesive (NOA81, Norland Products). The GRIN lens (NA 0.55, 64–519, Edmund Optics) was implanted for superficial imaging, or glued into the baseplate for deep-brain imaging, and the miniscope was mounted on a baseplate that was cemented to the skull.

## Cabling

To connect the NIN scope to the DAQ box, a thin (0.101 mm) coaxial wire (38awg A9438W-10-nd, Alpha Wire) of 50 cm length was attached to the miniscope on one end and connected through a connector set (ED90265-ND and ED8250-ND connectors, Digi-Key) to a thicker (1.17 mm FEP mm) coaxial wire (VMTX Mini,Pro Power) of 2 m at the other end. The 2 m wire was soldered to a 150 mm RG174 coaxial cable assembly with an SMA straight plug (2096227, LPRS) that could be directly attached to the DAQ board. The excitation LED driver was connected to the LED using 22 mm long ultra-thin wire (UT3607, Habia). A 25 mm long wire of the same type was connected to the optogenetic stimulus LED driver on the interface PCB. On one end a connector was attached (851-43-050-10-001000, Mill-Max of ED90265-ND, Digi-Key) to connect to an optogenetic probe.

## Acquisition and control software

Acquisition and control software was developed in the cross-platform Processing language (https://processing.org/). The software supports acquisition from up to two miniscopes and one USB webcam. The NINscope is controlled via a serial communication port using the Processing serial library, which supports custom microcontroller devices. The capture module in the video library for Processing version 2.0 was modified by changing YUV to a raw format to take advantage of the full 8-bit scale. To distinguish cameras and NINscopes of the same type a prefix is added to the string list of all attached devices. G4P, a Processing library (http://www.lagers.org.uk/g4p/) was used to create the software user interface, allowing access to the most common controls to change microscope

settings and update parameters such as light intensity, gain, black level, optogenetic stimulus duration, or G-sensor logging. Within the Processing sketch a capture event function transfers each captured image into a ring buffer for rapid processing of the image and to avoid long hard drive (HD) access times which could cause frame drops. A thread is used to save all images from ring buffer to HD. The size of the ring buffer can be modified in the sketch but is set to 60. The Last 4 pixels of a frame are reserved for a frame counter to monitor dropped frames. Every frame is saved as a gray-scale tiff image with the smallest possible header to reduce file size.

## Mice

We used both male and female C57BL/6 mice (weight range: 22–28-gram, age:>6 weeks). In experiments where Purkinje cells were optogenetically stimulated we used two transgenic male mice from a cross between the *Pcp2*-Cre Jdhu (JAX # 010536) and the Ai32 mouse line (JAX # 012569) on a C57BL/6 background. Mice were housed socially in groups of up to five mice prior to GRIN lens implantations, after which they were housed solitary. Mice were kept on a 12:12 hr light:dark cycle with lights on from 7:00 to 19:00. They received ad libitum access to water and food (Teklad, Envigo). Bedding material was provided for nest building. All performed experiments were licensed by the Dutch Competent Authority and approved by the local Animal Welfare Body, following the European guidelines for the care and use of laboratory animals Directive 2010/63/EU.

## GRIN lens implants and virus injections

Prior to surgeries mice were anesthetized with 3% Isoflurane before being transferred to a stereotactic apparatus after which anesthesia was maintained at 1.5% Isoflurane (flow rate: 0.3 ml/min $O_2$). For imaging of the cerebral and cerebellar cortex, a GRIN objective lens (1.8 mm diameter, 0.25 pitch, 64–519, Edmund Optics) was placed on the brain surface. A small incision was made in the skin after hair removal and disinfection of the skin with iodine solution (5%) and alcohol (70%). Lidocaine (100 mg/ml, Astra Zeneca, UK) was then applied to the exposed skull and the periosteum removed. The center coordinates for GRIN lens placement were located and a small ink dot was placed at the correct location relative to bregma (cerebellar Simplex lobule, AP: −5.8 mm ML: 2.2 mm; lobule VI, AP: −7.4 mm, ML: 0.0 mm; cortex, AP: 1.4 mm, ML: 1.5 mm). Coordinates were scaled relative to the mean bregma-lambda distance (of 4.21 mm) as specified in Paxinos mouse brain atlas. Prior to drilling of the bone, mice received i.p. Injections of 15% D-Mannitol in saline (0.55 ml/25gr) to aid diffusion of virus particles after virus injection. A 2 mm circular craniotomy was then drilled centered around the marked location. In between drilling the skull was kept moist with sterile saline. The skull flap and dura were then removed and virus (Cerebellum: AAV1.CAG.FLEX. GCaMP6f/AAV1.CMV.PI.Cre.rBG mixed 1:1 and diluted in saline 1:3; Cortex: AAV1.Syn.GCaMP6f. WPRE.SV40 diluted in saline 1:3, UPenn Vector Core) was injected at four locations. At each location 25 nl of virus was injected once at 350, twice at 300 and once at 250 μm depth at a rate of 25 nl/min with a Nanoject II Auto-Nanoliter Injector (Drummond Scientific Company, USA). The craniotomy was covered with gelfoam (Pfizer, USA) soaked in sterile saline (0.9% NaCl, B. Braun Medical Inc, USA). The GRIN lens was lowered using a vacuum holder placed in the stereotactic apparatus until the lens surface touched the brain and then lowered an additional 50 μm. The edges of the craniotomy were sealed with Kwik-Sil (WPI, USA). Dental cement (Super-Bond C and B, Sun Medical, Japan) was then applied around the lens to secure it in place. Kwik-Cast (WPI, USA) was used to cover and protect the lens. At the end of the surgery animals received an s.c. injection of 5 mg/kg Metacam.

To access deep-brain region dorsal striatum, GRIN relay lenses with a diameter of 0.6 mm (length: 7.3 mm, numerical aperture: 0.45, Inscopix, CA) were implanted (AP: 0.2, ML: 1.5, DV: −3.1). To ease implantation and reduce damage to tissue above dorsal striatum, a track was created using a 25G needle. Using a motorized stereotaxic arm, the needle was lowered slowly over the course of 10 min, left in place for 10 min, and then retracted with the same speed at a constant rate of 0.31 mm/min. Virus was delivered by lowering a Hamilton syringe over the course of 4 min to inject 500 nl of AAVdj.hSyn.GCaMP6s (N = 2 animals, *Figure 5*, N = 4 mice *Figure 6*) or AAV1.hSyn.GCaMP6f (N = 3 mice, *Figure 5*) into the dorsal striatum at a rate of 100 nl/min after which the syringe was left in place to increase diffusion of virus (5 min) and subsequently retracted (5 min). Subsequently, the GRIN relay lens was lowered (0.10 mm/min.) into the dorsal striatum (DV: −3.1). The gap between the GRIN relay lens and the skull was covered with cyanoacrylate glue and the lens was secured to

the skull using dental cement. The GRIN relay lens was covered and protected using Kwik-Cast (World Precision Instruments, USA). A 1.8 mm GRIN objective lens (64–519, Edmund Optics, UK) was secured onto the miniscope baseplate using cyanoacrylate glue and the miniscope was then mounted on the baseplate. Mice were head restricted using a custom-built running belt to allow locomotion, the scope with baseplate and objective GRIN lens mounted was then lowered above the GRIN relay lens until cells in the field of view became visible, the baseplate was secured to the skull with dental cement (coated with black nail polish) and the miniscope removed.

### LED probes

In experiments imaging from dorsal striatum, AAV5.hSyn.ChrimsonR.tdTomato was injected in either the orbitofrontal cortex (AP: 2.8, ML: 1, DV: −2.2) or secondary motor cortex (AP: 2.3, ML: 0.7 DV: −1.2). An LED probe, coated with biocompatible epoxy, designed to connect to NINscope with a red LED (645 nm peak emission, KingBright Inc, USA), was placed on the brain surface and the LED was secured to the skull using dental cement.

For cerebellar stimulation, a small 1 mm craniotomy was drilled above the cerebellar region of interest (Simplex lobule, AP: −5.8 mm, ML: 2.2 mm; Lobule VI, AP: −7.4 mm, ML: 0.0 mm; Crus II, AP: −7.3 mm, ML: + and −2.7 mm) following similar sterile procedures as described for the GRIN objective lens implant. The LED -covered in a layer of biocompatible epoxy- was inserted with proper orientation in the craniotomy, which was then sealed with a layer of Kwik-Sil (WPI, USA) followed by an additional layer of Kwik-Cast. The remaining probe wire and the probe connector were subsequently cemented to the skull with dental cement (Super-Bond C and B, Sun Medical, Japan).

### Imaging

Recordings from cerebellum and cerebral cortex were performed in the mouse home cage, while those obtained from dorsal striatum occurred in a custom open field arena (30 by 30 by 45 cm). The miniscopes were attached either following brief anesthesia (3% isoflurane induction, 30 min. post-anesthesia recovery time) to allow connection of the optogenetics LED driver connector to the LED probe, mounting of one or two scopes and focal adjustment, or -when using a modified baseplate or custom head bar- in a head-fixed condition where animals could run on a treadmill.

### Histology

Mice were transcardially perfused with 4% PFA in 1x PBS (10 mM $PO_4^{3-}$, 137 mM NaCl, 2.7 mM KCl). After fixation, brains were removed and kept overnight in 10% sucrose in PBS after which they were subsequently embedded in gelatin and left overnight in 30% sucrose in PBS. Brain sections (50 µm) were cut in a cryostat (LEICA CM3050 S), stained with DAPI and then mounted (Dako Fluorescence Mounting Medium, S3023). Confocal images were collected on a SP8 confocal microscope (Leica, Germany) and whole brain images were obtained by stitching multiple (1024 × 1024 pixel) acquisitions into a final image.

### Bead injections

To assess motion artifacts with dual NINscopes we made a 2 mm circular craniotomy, then injected a volume of 51 nl of 6 µm diameter fluorescent beads (InSpeck Green (505/515) Microscope Image Intensity Calibration Kit, 6 µm, Component G, ThermoFisher Scientific, USA) diluted to a concentration of 50% (corresponding to about 102 beads) into both cerebellum and cortex at a depth of 150–350 µm using a Nanoject II (see above) before retraction and prior to placing the objective GRIN lenses as described above. Animals were baseplated and imaged in the same way as for mice that had received virus injections. Shifts in the x-y plane were determined using the motion_metrics function in NoRMCorre.

### Calcium imaging analysis

Timestamps for all scopes were logged to disk and in case of dual scope recordings, frames of the videos were aligned prior to motion correction using NoRMCorre (*Pnevmatikakis and Giovannucci, 2017*) and signal extraction using CNMF-E (*Zhou et al., 2018*). We used the following parameters in CNMF-E to segment of Purkinje cell dendrites and cerebral cortical neurons respectively (Purkinje cell dendrites: gSig = 7, gSiz = 40; cerebral cortical neurons: gSig = 7, gSiz = 10, for both merge_thr

= [1e-1, 0.85, 0.5]). In the cerebellar stimulation experiments, neurons in cortex were classified as responders if the post-stimulus signal rose above the pre-stimulus mean+2σ. Onset times of these calcium transients were determined by fitting a sigmoid function to the transients and onset time was set to where the fitted function rose above mean+σ of the pre-stimulus baseline. For the dorsal striatum experiments, neurons were classified as responders if the signal during the stimulation period differed significantly from the pre-stimulus baseline signal. For every cell, activity pre-stimulus and during stimulation was averaged per trial and statistically tested using paired t-tests. Fiji (*Schindelin et al., 2012*) was used for raw data inspection and to create videos. Analyses were performed in Matlab (Mathworks, Nantucket), Python 3.7 (*Rossum, 1995*) and (*R Development Core Team, 2019*).

## Within and across-region synchronous pattern (SP) detection

Calcium transient events were inferred using a finite rate of innovation algorithm for fast and accurate spike detection (*Oñativia et al., 2013*) setting τ = 1 to obtain the onset times of calcium transients at a sub-recording rate resolution from the calcium transients per cell. These onset times were then convolved with an Epanechnikov kernel (steepness = 0.1) and summed over all cells, resulting in a kernel sum. All time intervals for which the kernel sum was two standard deviations above the mean were considered significant global synchronous events. If the onset times of two cells fell into the same synchronous event, these cells were considered to fire synchronously once. For all pairs of cells, we counted how many times these cells both fired inside the same synchronous event. Subsequently, we selected all synchronous pairs (SPs) that fired together at least five times.

These pairs were stored in a graph, where each node represents a cell and each edge the number of shared synchronous firing events. These graphs were converted to an arc diagram using the arc-diagram package (Gaston Sanchez, https://github.com/gastonstat/arcdiagram) in R. Cells were grouped by brain region (cerebellum, cortex) and sorted within the group by graph degree, that is the number of cells they correlate to. The degree of correlation is represented by the node radius. Custom-written scripts for SP extraction were written in Python and can be downloaded at: https://github.com/Romanovg185/sps-continuous-time-data.

## Analysis of behavioral acceleration

To distinguish whether accelerometer signals exceeded a mean+2σ threshold before or after an SP, we determined first where the largest mean signal occurred. We then selected for signals pre- or post-SP that rose above threshold. For the remaining signals we searched 300 ms around the SP (−150, +150 ms) for a rise above threshold. All other signals were classified as showing no SP-related change in behavioral acceleration.

## Analysis of rearing

We inspected video recordings of animal behavior to find approximate times of rearing (and to obtain the number of rearings per recording epoch) and then used the corresponding accelerometer data to determine rearing duration. For all rearings scored in our video data a clearly distinguishable rise and fall of the y and z accelerometer data could be discerned (cf. *Figure 2F*). The y acceleration channel was used to assess the start and end times of a rearing occurrence. For behavioral scoring different mice were used for all single and dual miniscope configurations.

## Design files and software availability

PCB and mechanical designs, firmware and acquisition software can be found on GitHub at: https://github.com/ninscope.

# Additional information

## Funding

| Funder | Grant reference number | Author |
| --- | --- | --- |
| Koninklijke Nederlandse Akademie van Wetenschappen | 240-840100 | Chris I De Zeeuw<br>Tycho M Hoogland |

| | | |
|---|---|---|
| H2020 European Research Council | ERC-2014-STG 638013 | Ingo Willuhn |
| Nederlandse Organisatie voor Wetenschappelijk Onderzoek | 2015/06367/ALW 864.14.010 | Ingo Willuhn |
| Nederlandse Organisatie voor Wetenschappelijk Onderzoek | ALWOP.2015.076 | Chris I De Zeeuw Tycho M Hoogland |
| H2020 European Research Council | ERC-adv ERC-POC | Chris I De Zeeuw |
| Topsector Life Sciences & Health (LSH) | LSHM18001 | Tycho M Hoogland |

The funders had no role in study design, data collection and interpretation, or the decision to submit the work for publication.

## Author contributions

Andres de Groot, Conceptualization, Resources, Data curation, Software, Formal analysis, Validation, Methodology, Writing - original draft, Writing - review and editing; Bastijn JG van den Boom, Conceptualization, Data curation, Formal analysis, Validation, Investigation, Visualization, Writing - original draft, Writing - review and editing; Romano M van Genderen, Conceptualization, Resources, Software, Formal analysis, Methodology, Writing - review and editing; Joris Coppens, Conceptualization, Formal analysis, Validation, Methodology, Writing - review and editing; John van Veldhuijzen, Joop Bos, Conceptualization, Resources, Methodology; Hugo Hoedemaker, Resources, Validation, Investigation, Methodology, Writing - review and editing; Mario Negrello, Supervision, Methodology, Writing - review and editing; Ingo Willuhn, Supervision, Funding acquisition, Writing - review and editing; Chris I De Zeeuw, Conceptualization, Funding acquisition, Writing - review and editing; Tycho M Hoogland, Conceptualization, Resources, Data curation, Formal analysis, Supervision, Funding acquisition, Validation, Investigation, Visualization, Methodology, Writing - original draft, Project administration, Writing - review and editing

## Author ORCIDs

Bastijn JG van den Boom (iD) http://orcid.org/0000-0002-0853-3763
Mario Negrello (iD) https://orcid.org/0000-0002-8527-4259
Ingo Willuhn (iD) http://orcid.org/0000-0001-6540-6894
Chris I De Zeeuw (iD) https://orcid.org/0000-0001-5628-8187
Tycho M Hoogland (iD) https://orcid.org/0000-0002-7444-9279

## Ethics

Animal experimentation: All performed experiments were licensed by the Dutch Competent Authority and approved by the local Animal Welfare Body, following the European guidelines for the care and use of laboratory animals Directive 2010/63/EU.

## Decision letter and Author response

Decision letter https://doi.org/10.7554/eLife.49987.sa1
Author response https://doi.org/10.7554/eLife.49987.sa2

# Additional files

## Supplementary files

• Transparent reporting form

## Data availability

Hardware, firmware and software have been deposited at GitHub under an MIT license.

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
