## [Decision Letter]

**Acceptance summary:**

The reviewers and editors appreciated the versatility and practical usability of the openly available NINscope system. Compared to other systems, NINscope incorporates advantages in weight and size that allow for coupling with accelerometers, optogenetic stimulation, and the possibility for dual site imaging. Together, these open up broader possibilities than exist with other systems. Because it is being made openly available, we expect this system will be a particularly useful addition for labs that don't have the ability to develop these additional features on their own through modification of existing systems.

**Decision letter after peer review:**

Thank you for submitting your article "NINscope: a versatile miniscope for multi-region circuit investigations" for consideration by *eLife*. Your article has been reviewed by three peer reviewers, and the evaluation has been overseen by a Reviewing Editor and Kate Wassum as the Senior Editor. The following individual involved in review of your submission has agreed to reveal their identity: Denise J Cai (Reviewer #1).

The reviewers have discussed the reviews with one another and the Reviewing Editor has drafted this decision to help you prepare a revised submission.

Summary:

Your paper generated quite a lot of discussion in the consultation phase. On the one hand, the reviewers felt that the individual technical advances here were relatively small. On the other hand, there was agreement that in principle, the small advantages in weight and size, although incremental individually, when coupled with accelerometers, optogenetic stimulation, and possibility for dual site imaging, could together open up broader possibilities than exist with other systems. The main advance of the NINscope was seen as potentially residing in versatility and practical usability rather than in technical specs per se – highlighting HOW to combine imaging with this smaller footprint with other applications (opto, dual imaging) is a nice contribution. If this system were made openly available, it might be a useful addition for labs that don't have the ability to develop these additional features on their own through modification of existing systems.

As a Tools and Resources article, there is no requirement to report major new biological insights or mechanisms, but it must be clear that they will enable such advances to take place. Since the goal here is not to answer biological questions, it needs to be made more clear that the experiments are proof-of-principle (i.e. the scope and associate technology works) demonstration experiments AND the data need to be high quality.

Key issues were raised that would need to be addressed to conclusively demonstrate these points, summarized here:

1) Most importantly, there are concerns about the quality of the imaging data as represented in the supplemental videos, including apparent z-axis motion artifact, deconvolution of GCaMP signals, and Purkinje cell dendrite segmentation (touched on in varying ways by all 3 reviewers, but especially reviewer 3). The videos are also highly processed, with red flashes indicating activity, obscuring the segmentation and signal extraction. To ensure that the smaller footprint of the NINscope does not sacrifice stability and image quality, we ask you to:

– Provide YFP (or other static fluorophore) controls for motion artifacts. This is particularly (but not only) relevant for the instances of increased activity across regions that correlates with head acceleration.

– Provide more raw data and information about segmentation. In the videos, please provide the same data calcium imaging data in raw form, after motion correction, and then again after segmentation. This could be done by showing the same video for these three steps.

2) There was concern that too much is being made in terms of biological findings resulting from the disparate proof-of-principle experiments that are shown. Those studies were seen as underpowered, lacking experimental detail and lacking control experiments to make strong biological conclusions. Moreover, answering all these biological questions in detail is outside the scope of the tools and resource paper. Please focus on the tools and methods aspects (and explicate on that) and temper the conclusions on findings of coordinated neural activity across brain regions.

3) To validate the utility of mounting 2 scopes, in behaving animals please provide

– more revealing images of the dual mounted configuration (as reviewer 1 suggested)

– quantification of behavior with 0,1,2 scopes mounted (suggested by reviewer 2).

4) To enable straightforward evaluation of the specs of NINscope compared with other existing options (UCLA miniscope, Inscopix), please include a comparison table as suggested by reviewer 2.

5) Please provide validation of the true decoupling of stimulation and recording site for optogenetics (Reviewer 3 point 2).

We therefore invite you to submit a revised manuscript that fully addresses these concerns, as well as the full set of reviewers' comments, below.

Reviewer #1:

Groot et al. report the design and application of a novel miniscope – NINscope – whose primary advance over existing miniscope technologies is its small and light-weight nature, which allows implantation of two of them for simultaneous imaging across two brain regions in mice. Further, NINscope is capable of integration with optogenetics, for concurrent imaging and optogenetic manipulation of brain circuits. The authors demonstrate that NINscope is capable of acquiring high resolution videos whose underlying neural activity can be deconvolved using conventional data processing techniques (i.e. CNMF). They then discover synchronous activity patterns (SPs) across cortex and cerebellum during onset of motion, using NINscope to record across the two regions and an integrated accelerometer to measure behavioral motion. To demonstrate that NINscope is amenable to integration with optogenetics, the authors then image cortical calcium activity while stimulating Purkinje cells of 4 cerebellar subregions and show that stimulation of each subregion increases cortical calcium activity. Moreover, assessment of behavioral acceleration suggests that stimulation of one hemisphere of Crus II leads to head movement in the opposite direction. To demonstrate NINscope's capacity to image from deep brain structures, the authors implanted a GRIN lens in the dorsal striatum using a relay lens system and recorded striatal neuron calcium activity as mice explored an open field. Using this system, the authors observe individual neurons tuned to turns in the direction opposite the recorded hemisphere. Finally, the authors use deep brain imaging and optogenetics to demonstrate that optogenetic stimulation of OFC and M2 differentially influence striatal neuronal activity. Overall, NINscope serves as a valuable and flexible tool that allows investigation of neuronal population dynamics across brain regions. Because of its small footprint, it allows for flexible use such as combining it with additional NINscope or optogenetics. This tool will surely benefit the neuroscience community broadly, however, some additional details of methodology and clarification of certain areas (addressed below) would improve this manuscript greatly. And while it is commendable that the authors show several examples of how NINscope can be used experimentally, it is unclear what the biological discovery is across these different experiments which seem to be tackling disparate research questions. The authors may want to more clearly state what the hypotheses are based on prior research and what findings are consistent with prior research and which are novel or challenge prior views. I understand this may increase the scope of this paper too broadly, which I would like to suggest that the authors consider focusing their experimental questions.

Reviewer #2:

de Groot et al. present a new open-source miniature fluorescent microscope. In direct comparison to the most widely used miniature fluorescent microscopes, the new scope (dubbed NINscope) has a few additional features/properties that will make it appealing for some groups to adopt. In particular, the NINscope has a smaller footprint, is lighter, and contains on-board accelerometers and LED drivers to control external LEDs. These features are well-integrated into the design and their utility is demonstrated in the manuscript. To demonstrate functionality of this new microscope and its associated hardware and software, the authors confirmed multiple previous findings from the movement literature. Namely, the authors use NINscope to successfully track activity of purkinje cell dendrites and dorsal striatum neurons during movement, to examine functional connectivity between cerebellum and cortex, and to confirm findings that orbitofrontal and motor cortex provide top-down control of dorsal striatum. In my eyes, the primary deliverable of this paper is the microscope itself and very little more is needed to demonstrate its function and utility. Some modifications, however, would strengthen the dissemination of relevant information about the NINscope. The neurobiological experiments are perhaps underpowered, but not in a way that undermines the microscope itself. For example, the known limitations of detecting a decrease in signal from GCaMP, might provide some bias in Figure 6, but this possibility is not detracting from the technical advances made here.

One addition that would strengthen this manuscript is a more detailed comparison between the NINscope and other open-source (maybe commercial too when applicable) miniature microscopes. This could be in the form of text or figure, but a comprehensive table might be the most succinct manner of conveying information that would inform future users. I recognize that the last author has a review on this topic, but that review does not capture the NINscope itself or much detail on the wire-free scope from Shuman et al. (bioRxiv, 2018). Direct comparisons of weight, footprint size, control schemes, focusing mechanism, software considerations, accessory hardware, and stimulation ability are just some of the variables/limitations that might sway a user from one scope to another.

The other main comparison that would strengthen claims made about the scope hardware itself (should be gathered easily or from existing data) is a quantified comparison of locomotor behavior with zero, one, and two NINscopes headmounted. The authors suggest normal behavior is seen and refer to cage exploration, grooming, eating, and rearing, but these are not actually demonstrated. Rearing epoch is defined by video and IMU data in 2G, presumably number of rears could be collected from one- and two-scope setups and compared.

Including more information about the LEDs for optical stimulation would be helpful. These appear to be very simple applications of standard LEDs and it might be useful for others to employ these with or without the NINscope. Supplementary photos and/or diagrams of these would help.

Light power for imaging was not reported for Figure 6. Was it the same as Figure 5? Due to the possibility of activating the blue tail of ChrimsonR, please include this information.

Reviewer #3:

In this manuscript by de Groot et al., the authors develop and characterize a miniature epifluorecent microscope to be used in freely behaving rodents for calcium imaging. The microscope is similar to previously published miniscopes (UCLA miniscope, Inscopix nVista, and the Finchscope) with a few minor differences. Namely, it is slightly lighter than the Inscopix scope, and contains and accelerometer/LED for positional tracking. It also has an extra LED that can be used for photostimulation. While other systems do not offer this, it seems like it would be easily implemented with other systems if needed.

Specific comments:

1) One of the major points made throughout the manuscript is that this system permits for multisite recording. While this is demonstrated in the cerebellum and frontal cortex, it is unlikely this could be achieved in many other regions. The frontal cortex and cerebellum are quite far apart (thus while it is feasible in this example), but this will be highly limited to other duel sites.

2) The optogenetic stimulation coupled with imaging is interesting, and has been previously reported using the Inscopix nVoke system (where stimulation and recording occur in the same field of view). The advantage in this paper is that stimulation can be decoupled from the recording site. While this is interesting, it does not appear that control experiments where mice were not expressing ChR2 were performed. Thus, it is possible that neural activity evoked by stimulation could be due to some sort of heating or light induced artifact.

3) I have some concerns with the data quality presented in the supplementary videos. Many of them show non-correctable z-motion artifact, and the cerebellar dendritic recordings seem to show a lot of out of focus fluorescence which will make it difficult to extract signals from single dendrites. The authors used the recently published CMNF-E algorithm for signal extraction, which is extracting something from the data, but it is unclear whether extracted signals actually represent bona vide single dendrites.

---

## [Author Response]

Reviewer #1:[…] Overall, NINscope serves as a valuable and flexible tool that allows investigation of neuronal population dynamics across brain regions. Because of its small footprint, it allows for flexible use such as combining it with additional NINscope or optogenetics. This tool will surely benefit the neuroscience community broadly, however, some additional details of methodology and clarification of certain areas (addressed below) would improve this manuscript greatly. And while it is commendable that the authors show several examples of how NINscope can be used experimentally, it is unclear what the biological discovery is across these different experiments which seem to be tackling disparate research questions. The authors may want to more clearly state what the hypotheses are based on prior research and what findings are consistent with prior research and which are novel or challenge prior views. I understand this may increase the scope of this paper too broadly, which I would like to suggest that the authors consider focusing their experimental questions.

We thank the reviewer for their time to carefully read through our manuscript, valuable suggestions to improve it, and overall positive assessment. We agree with the conclusion that the experiments demonstrated are not necessarily connected with an overarching hypothesis, but our goal for this Tools and Resources manuscript was above all to show NINscope’s versatility and applicability, which has led to a disparate set of experiments. Acknowledging the above, we have now toned down the conclusions drawn with regard to the biological findings.

Reviewer #2:[…] Some modifications, however, would strengthen the dissemination of relevant information about the NINscope. The neurobiological experiments are perhaps underpowered, but not in a way that undermines the microscope itself. For example, the known limitations of detecting a decrease in signal from GCaMP, might provide some bias in Figure 6, but this possibility is not detracting from the technical advances made here.One addition that would strengthen this manuscript is a more detailed comparison between the NINscope and other open-source (maybe commercial too when applicable) miniature microscopes. This could be in the form of text or figure, but a comprehensive table might be the most succinct manner of conveying information that would inform future users. I recognize that the last author has a review on this topic, but that review does not capture the NINscope itself or much detail on the wire-free scope from Shuman et al. (bioRxiv, 2018). Direct comparisons of weight, footprint size, control schemes, focusing mechanism, software considerations, accessory hardware, and stimulation ability are just some of the variables/limitations that might sway a user from one scope to another.

Thank you for this suggestion. We have added a Table in which we compare NINscope to other currently available miniscopes (open-source and commercial).

The other main comparison that would strengthen claims made about the scope hardware itself (should be gathered easily or from existing data) is a quantified comparison of locomotor behavior with zero, one, and two NINscopes headmounted. The authors suggest normal behavior is seen and refer to cage exploration, grooming, eating, and rearing, but these are not actually demonstrated. Rearing epoch is defined by video and IMU data in 2G, presumably number of rears could be collected from one- and two-scope setups and compared.Including more information about the LEDs for optical stimulation would be helpful. These appear to be very simple applications of standard LEDs and it might be useful for others to employ these with or without the NINscope. Supplementary photos and/or diagrams of these would help.

This is an excellent suggestion. Per the reviewer’s suggestion we have quantified the number of rearings in mice wearing single and dual miniscopes (CBL-CTX configuration) for 4 mice. We find that rearing frequency and duration is unaffected by adding an additional miniscope.

We have updated Figure 1 in that we now include a separate panel (1D) to show the LED probe we built to use in combination with NINscope, and added a short paragraph to the text describing its parts. The assembly of probes is described in detail on our GitHub hardware wiki: https://github.com/ninscope/Hardware/wiki/8.-LED-probe.

Light power for imaging was not reported for Figure 6. Was it the same as Figure 5? Due to the possibility of activating the blue tail of ChrimsonR, please include this information.

We added the information on light power as requested. The reviewer raises a valid point about the possibility that ChrimsonR could be activated by blue light. We believe activation of the blue (excitation) tail of the ChrimsonR is likely mitigated by the following factors:

1) The light power used for striatal imaging as measured under the GRIN relay lens -now reported in Figure 2—figure supplement 1 corresponds to about 170 µW. This light power is more than an order of magnitude lower than the light power of the red LED used to stimulate the OFC and M2 in cortex.

2) We performed these experiments four weeks post-injection when terminal expression of ChrimsonR is still relatively weak.

3) Even if blue light excites terminals expressing ChrimsonR we still observe a differential postsynaptic response upon stimulation of M2 and OFC.

Reviewer #3:In this manuscript by de Groot et al., the authors develop and characterize a miniature epifluorecent microscope to be used in freely behaving rodents for calcium imaging. The microscope is similar to previously published miniscopes (UCLA miniscope, Inscopix nVista, and the Finchscope) with a few minor differences. Namely, it is slightly lighter than the Inscopix scope, and contains and accelerometer/LED for positional tracking. It also has an extra LED that can be used for photostimulation. While other systems do not offer this, it seems like it would be easily implemented with other systems if needed.

The points made above highlight that perhaps we should have been more explicit about the effort that goes into designing and building integrated circuit boards to allow in-software control of all aspects of the experimental recordings. We designed and built two compact integrated circuit boards for NINscope while reducing the miniscope weight, integrating an accelerometer, adding dual-site imaging capabilities and permitting optogenetic stimulation.

Adding functionality by way of separate components, would add unnecessary weight or cables, which might impact animal behavior and above all would require time-consuming steps to ensure post-hoc data synchronization. Even if these features could be added it would require specific technical skills that not everyone has access to.

We believe that by sharing this resource as an open-source tool we can ensure that a larger group of users can benefit from its specific features and allow them to address novel research questions.

Specific comments:1) One of the major points made throughout the manuscript is that this system permits for multisite recording. While this is demonstrated in the cerebellum and frontal cortex, it is unlikely this could be achieved in many other regions. The frontal cortex and cerebellum are quite far apart (thus while it is feasible in this example), but this will be highly limited to other duel sites.

Thank you for pointing this out. In order to more clearly demonstrate the suitability of NINscope for multi-site recordings, we now demonstrate additional dual scope configurations in Figure 3—figure supplements 1 and 2, including bi-hemispheric imaging from somatosensory and visual cortex, hippocampus, or two cerebellar hemispheres.

2) The optogenetic stimulation coupled with imaging is interesting, and has been previously reported using the Inscopix nVoke system (where stimulation and recording occur in the same field of view). The advantage in this paper is that stimulation can be decoupled from the recording site. While this is interesting, it does not appear that control experiments where mice were not expressing ChR2 were performed. Thus, it is possible that neural activity evoked by stimulation could be due to some sort of heating or light induced artifact.

We now include Figure 4—figure supplement 1, which demonstrates that stimulation at multiple cerebellar locations neither triggers a response in neurons of cortex nor a behavioral response.

3) I have some concerns with the data quality presented in the supplementary videos. Many of them show non-correctable z-motion artifact, and the cerebellar dendritic recordings seem to show a lot of out of focus fluorescence which will make it difficult to extract signals from single dendrites. The authors used the recently published CMNF-E algorithm for signal extraction, which is extracting something from the data, but it is unclear whether extracted signals actually represent bona vide single dendrites.

We agree with the reviewer that the visualization in the videos as presented was unclear. The red flashes represented mean-subtracted raw fluorescence superimposed on the raw data and the videos were saved in a compressed format, which did not clearly represent the data. We therefore now provide higher quality supplemental videos that include the step-by-step process from raw data to motion correction to segmentation throughout the manuscript. These videos in combination with dual miniscope experiments in which we injected fluorescent beads into cerebellum and cortex reveal that motion artefacts are actually fairly modest.

To show that CNMF-E can successfully segment Purkinje cell dendrites we have updated Figure 2 and other figures to show the extracted spatial footprints. Our spatial footprints of Purkinje cell dendrites meet the criteria of parasagittally aligned elongated structures and the extracted signals match what we expect from complex spike evoked calcium transients having firing rates of around 1Hz. Moreover, given the anatomy of the cerebellum and the fact that we are imaging from the surface of the cerebellar cortex in combination with Purkinje cell selective transduction of GCaMP6f we are confident that we are reporting signals from individual Purkinje cell dendrites.